



# Origin of Great Unconformity Obscured by Thermochronometric Uncertainty

Matthew Fox[1], Adam G.G Smith[1], Pieter Vermeesch[1], Kerry Gallagher[2], Andrew Carter[3]

[1] London Geochronology Centre, Department of Earth Sciences, University College London, Gower St., London, WC16BT, UK

[2] Géosciences Rennes/OSUR, University of Rennes, Rennes, France

[3] London Geochronology Centre, Department of Earth Sciences, Birkbeck College, Gower St., London,

WC16BT, UK

*Correspondence to*: Matthew Fox (m.fox@ucl.ac.uk)

**Abstract.** Thermochronology provides a unique perspective on the magnitude of rock that is eroded during, and the timing of, unconformities in the rock record. Recently, thermochronology has been used to reinvigorate a long-standing debate about the origin of the Great Unconformity, a global

erosional event that represents a time period of almost a billion years at the end of the Precambrian. The (U-Th)/He in zircon system is particularly well suited to provide this perspective because it is very sensitive to long durations of time at relatively low temperatures (< 200-250°C). However, the diffusion kinetics of $^4$He in zircon change dramatically as a result of radiation damage to the crystal lattice. Therefore, our ability to resolve thermal histories is fundamentally limited by how well we

know parameters controlling helium diffusion and their uncertainties. Currently, there is no estimate of how these uncertainties impact the inferred thermal histories. Here we determine uncertainties in the Zircon Radiation Damage and Annealing Model (ZRDAAM, Guenthner et al. 2013) that describes changes in $^4$He diffusion kinetics as a function of radiation damage. We show that the dispersion in predicted zircon (U-Th)/He ages for a given thermal history can be 100s Ma for a specific amount of

radiation damage and that thermal histories are less well resolved than previously appreciated. Additional diffusion experiments and calibration with natural laboratories would provide better constraints on diffusion kinetic parameters.

## 1. Introduction

Thermochronometry is widely used to constrain the evolution of Earth's surface and upper crust, transforming our understanding of the magnitudes of sedimentation and erosion linked to climate and tectonic change. Notably, thermochronometry has been used recently to understand the development of



the Great Unconformity (DeLucia et al., 2018; Keller et al., 2019; Flowers et al., 2020; McDannell et al., 2022), a global feature that marks the boundary between the Precambrian and Phanerozoic. The

time-period that the Great Unconformity spans varies depending on location, but on the North American craton, in the Grand Canyon, the erosional event spans from circa 1200 to 250 million years (Thurston et al., 2022). The origin of the Great Unconformity has been debated for over 125 years, with recent contributions using thermochronometry to gain new insight (McDannell et al., 2022). This approach highlighted that erosion rates increased across the North American craton during

Neoproterozoic glaciation, supporting the hypothesis that the Great Unconformity is the result of glacial erosion (Keller et al., 2019), as opposed to diachronous tectonic events (Flowers et al., 2020).

The principle behind thermochronometry is that rocks experience temperature changes over geological timescales: rocks closer to Earth's surface are cooler than deeper ones and so exhumation leads to

cooling. The time scales associated with these changes in temperature are determined using the concepts of geochronology. The daughter products of radioactive decay used for thermochronometry have a temperature dependent rate of loss from the target mineral. At high temperatures the daughter products are effectively lost instantaneously. By contrast, as the rock cools to lower temperatures, the rate of diffusive decreases and daughter products are progressively retained until there is effectively no

diffusive loss. Therefore a measured age reflects the duration of residence at low temperature (Dodson, 1973; Zeitler et al., 1987; Reiners, 2005).

One of the most widely used methods for deep-time thermochronometry is (U-Th)/He in zircon (ZHe) (Reiners, 2005). The basis of this method is that the radioactive elements uranium and thorium are

incorporated into crystals of zircon and the decay product, helium, is trapped in the crystal lattice. Helium diffuses from zircon at high temperatures, but is retained at lower temperatures. The exact temperature range at which the transition from closed-system to open-system behaviour occurs is dependent on damage to the crystal lattice produced primarily from recoil during decay processes (Guenthner et al., 2013). Radiation damage accumulates at a rate that depends on the amount of

uranium, thorium, and samarium in the crystal. This means that two zircon crystals from the same rock, experiencing the same thermal history, could have very different thermal sensitivity. In turn, models accounting for this variable temperature sensitivity as a function of radiation damage can be used to leverage more complex thermal histories than constant kinetic parameter models. This approach has successfully been used to infer deep-time erosion rate histories, and in the case of the Great



Unconformity, has reinvigorated debate on its origin (Flowers et al., 2020; Peak et al., 2021;

McDannell et al., 2022)

Both those in favour of a glacial (McDannell et al., 2022) and tectonic (Flowers et al., 2020) origin of

the Great Unconformity have interpreted ZHe data using the Zircon Radiation Damage And Annealing

Model (ZRDAAM) of Guenthner et al. (2013). In this model, a crystal is composed of undamaged and

damaged parts that combine to give bulk diffusion kinetics as a function of the amount of radiation

damage. Accumulated radiation damage is calculated based on the concentrations of the parent

elements and the thermal history of a sample. The damage accumulates at low temperatures, but can be

annealed at higher temperatures, calculated using fission track annealing kinetics (Yamada et al., 1995;

Rahn et al., 2004; Tagami, 2005; Yamada et al., 2007). Therefore, the diffusivity of helium at a specific

temperature is a function of the past thermal history. This makes the overall problem very non-linear so

that changing the temperature at some time in the thermal history can have unexpected effects on the

resulting age, as also shown for the (U-Th)/He in apatite system by Fox and Shuster (2014).

Using the radiation damage and annealing model (RDAAM) with inverse models, researchers have

resolved tight temperature constraints on thermal histories over billion year timescales. For example,

Thurston et al. (2022) inferred a 1700 Ma thermal history from ZHe ages in Eastern Grand Canyon.

Parts of this history were reported to within less than 10 degrees between 700 and 250 Ma and then

again from 15 -7 Ma. It is unclear whether the data really provide such tight constraints on

temperatures in the past or whether these are at least partly the consequence of model assumptions

and/or, potentially, overconfidence in the adopted diffusion kinetic parameters.

Calibration of ZRDAAM has been carried out using measured diffusion kinetics of crystals with known

amounts of radiation damage (Guenthner et al., 2013). However, the accuracy and precision of this

model has not been assessed. In particular, it is unclear how the propagation of uncertainties to model

parameters affects the dispersion or sensitivity of predicted thermochronometric ages. Here we show

that the uncertainties in the radiation damage model make it challenging to accurately infer the timing

and magnitude of unconformities in the deep past. We begin by highlighting why ZRDAAM needs to

be calibrated accounting for uncertainties and present our new calibration. We then propagate

uncertainties from this model calibration through time temperature paths. We show that natural

variability in radiation damage annealing parameters causes ZHe ages to be very dispersed even for

crystals of the same size and radiation damage levels. Using QTQt (Gallagher, 2012), we show that



different diffusion kinetics can lead to the onset of cooling for resolved thermal histories from inverse methods varying by hundreds of millions of years.

**2. The existing calibration of the radiation damage and annealing model**

The rate of diffusion is controlled by the diffusivity and the curvature of the concentration of the diffusant (Fick's Law). Although the production distribution of the diffusant (helium in our case) can be important in some scenarios, diffusion tends to smooth the distribution. More significant is the fact that the diffusivity can vary by orders of magnitude with variations in temperature. The diffusivity at

any temperature is given by the Arrhenius equation of the diffusivity D (cm$^2$/s):

$$D(T) = D_0 e^{\frac{-E_a}{RT}}$$

(1)

where $D_0$ is the frequency factor (cm$^2$/s), $E_a$ is the activation energy (kJ/mol), $R$ is the gas constant

(J/K/mol) and $T$ is the temperature in Kelvin. Taking the logarithm of equation 1, gives:

$$ln(D(T)) = \frac{-E_a}{R}\frac{1}{T} + ln(D_0)$$

(2)

so that the slope of the line between ln($D(T)$) and 1/$T$ gives $E_a$/$R$ and the intercept of the line provides ln($D_0$). Diffusion experiments in which a crystal is step-wise degassed *in vacuo* are used to calculate

*D(T)* for different combinations of specific temperatures and time (Fechtig and Kalbitzer, 1966). The resulting plot can be used to determine the Arrhenius parameters ($D_0$, $E_a$). Importantly, estimates of the two model parameters extracted from this linear inversion covary with one another, i.e., ln($D_0$) is strongly correlated with $E_a$.

Analyzing different crystals with known radiation damage values allows us to assess how diffusion kinetics vary with damage. It is challenging to visualize both model parameters ($D_0$ and $E_a$) for each crystal as a function of radiation damage and so it is common to combine the parameters and calculate a diffusivity at a specific temperature or a closure temperature at a specific cooling rate. By combining the parameters, however, information on how the two parameters are correlated is lost.


The results of Guenthner et al., (2013)'s diffusion experiments highlight two general trends. At low damage values the closure temperature increases with increasing damage. At higher damage values, the closure temperature decreases with increasing damage. This general behaviour has been reproduced in





numerical models conducted at a range of scales (Ketcham et al., 2013; Gautheron et al., 2020). To

interpret these trends in ZRDAAM, a model is used in which the diffusion kinetics for a specific

radiation damage value are a combination of a theoretical minimally damaged crystal and an extremely

damaged crystal. The diffusion kinetics of these end-member crystals need to be estimated.  The

frequency factor of the minimally damaged crystal ($^zD_0$) was estimated by extrapolating the frequency

factors of measured crystals down two orders of magnitudes using a power-law relationship. The

activation energy for theoretical crystal ($^zE_a$) is set as the average of the activation energies of

minimally damaged crystals (see Guenthner et al., 2013 for details). Extrapolating values to a

minimally damaged crystal, however, will add uncertainties and there is no obvious way to account for

these in the power-law relationship. Crucially, this approach does not account for the correlations

between the model parameters. This is important because the correlations provide additional

information that can yield more precise estimates of model parameters and allow propagation of

uncertainties into model predictions. The diffusion kinetics for the extremely damaged crystal are

estimated using sample N17 (Guenthner et al., 2013), and also involve correlated model parameters

($^{N17}D_0$ and $^{N17}E_a$). However, the accuracy of this model has only been assessed by looking at general

trends in model predictions. Here we attempt to formally quantify the uncertainty in model parameters

and how these uncertainties translate to uncertainties in temperature sensitivity of the ZHe system, and

in particular predicted ZHe ages.

**3. A new calibration of the zircon radiation damage and annealing model**

In order to account for the correlation between the frequency factor and the activation energy, we

model the measured helium diffusivities directly. We use the same diffusion data and parameterisation

as Guenthner et al. (2013). However, in contrast with Guenthner et al. (2013), we determine the

diffusion of the end member crystals using the radiation damage model directly, rather than by non-

linear extrapolation from high to low radiation damage levels. Our goal is not to simply increase the

accuracy of the model parameters but to determine their precision. By tracking correlations in model

parameters, we can simulate (U-Th)/He in zircon ages accounting for uncertainties in the original

diffusion experiments.

The model we fit is given by equation 8 of Guenthner et al. (2013) and describes the diffusivity $D$ as a

function of the amount of damage:





$$\frac{1}{\frac{D}{a^2}} = \frac{f_c'}{\left(\frac{l_{int0}}{l_{int}}\right)^2 * \left(\frac{D_z}{(a * f_c')}\right)} + \frac{f_a'}{\frac{D_{N17}}{(a * f_a')^2}}$$

(4)

where $f_c'$ and $f_a'$ are the crystalline and amorphous fractions, respectively, $l_{int}$ and $l_{int,0}$ are parameters

describing how far a helium atom can travel within a crystal lattice without encountering damage in a

damaged crystal and an undamaged crystal respectively, and $a$ is the grain size. $D_z$ and $D_{N17}$ can be

calculated using the diffusion kinetics of the undamaged and damaged theoretical crystals, using

equation 4. Therefore, for every sample with a known amount of damage, we can calculate different

diffusivity values for degassing steps using model parameters $^{N17}D_0$ and $^{N17}E_a$ for $D_{N17}$ and $^zD_0$ and $^zE_a$

for $D_z$. We use the Bayesian Markov Chain Monte Carlo (MCMC) method incorporating the

Metropolis-Hastings algorithm to sample the full posterior distribution of the model parameters. We

tune the proposal distributions to ensure that approximately 20% of the proposed models are accepted

as this represents an efficient balance between exploring parameter space and sampling the parameter

values. The Markov Chain is initialised with the model parameters of Guenthner et al. (2013) and the

algorithm runs until 1 million sets of model parameters have been accepted. The likelihood function is

defined as a least squares fit to the data. However, the degassing experiments of Guenthner et al. (2013)

each have different numbers of steps. The likelihood function involves a summation, related to the

number of steps, and so number of data points in each experiment, experiments with more steps would

tend to dominate our results. Similarly, key experiments which might have fewer steps would have far

less influence on the model parameter estimation. To account for this problem, the misfit for each

experiment is weighted accordingly, so that the log-likelihood (LL) function is:

$$LL = 0.5 * \sum_{i=1}^{N} \frac{1}{M_i} \sum_{j=1}^{M_i} \left(\frac{D_{i,j} - P_{i,j}}{\sigma}\right)^2$$


(5)

where $N$ is the number of crystals analyzed, $M_i$ is the number of degassing steps used in the inversion

for a specific crystal, $D_{i,j}$ is the observed diffusivity for a specific crystal at a specific degassing step, $P_{i,j}$

is corresponding predicted diffusivity calculated with Equation 4, and $\sigma$ is the estimated uncertainty

set to 1 ln(1/s) here, based on reported uncertainties. Preliminary experiments highlighted that the

sampling was relatively insensitive to the diffusion kinetics of the damaged N17 crystal. To ensure that



the diffusion kinetics of this unique and crucial end member crystal was accurately captured, we reduced the uncertainty of the diffusion data for N17 to 0.1 ln(1/s).

The comparison of the predicted and observed diffusivities for the best fitting model is shown in Figure
1B. We also show the model fit with the original Guenthner et al., (2013) parameters in Figure 1A. Results of our analysis are plotted as 4 histograms showing the original model parameters and our inferred model parameters (Figure 2). The posterior probability of the model parameters is proportional to the height of the model histograms. For the amorphous crystal, our maximum a posteriori model parameter values are close to the original values, shown in red. However, for the low-damaged crystal,
the model predictions are quite different, although we note that the original values are close to the 2D probability peak (Figure 3) which accounts for model correlations. Additionally, we will focus on the overall spread in the values of these model parameters and we explore the importance of this variability in the next section.

**4. Propagating model uncertainties**

To assess the importance of the uncertainties in the radiation damage and annealing model parameters, we predict ages using a simple thermal model. The time-temperature path is chosen to resemble that of the Minnesota samples from McDannell et al. (2022), and represents a typical inferred time temperature path of a Deep Time target locality. Here the rocks have been below 600°C since 1.5 Ga.
From 700 Ma to 650 Ma the rocks cooled from 200 to 150°C, and then gradually to 0°C by 200 Ma before experiencing reheating at 50 Ma to 100°C. Between 50 Ma and the present the rocks cooled linearly to 0°C. Radiation damage accumulates throughout this history such that some zircon crystals that transitioned from open to closed system behaviour during the cooling event at 700 Ma, transitioned
back to open behaviour simply due to the accumulation of radiation damage. Some crystals however, with intermediate temperature sensitivity only record the final cooling event. Other crystals, with low-temperature sensitivity, also record cooling associated with the 100 Ma burial event. In terms of constraining thermal history models, this potential to have a wide range of temperature sensitivities within a single sample makes the ZHe method very powerful.


To calculate thermochronometric ages we use the radiation damage and annealing model of Guenthner et al. (2013) with our updated model parameters. Note, this implementation of the model has been used previously (Tripathy-Lang et al., 2015). 20 different crystals are simulated spanning an effective U





concentration ([eU] = [U]+0.24[Th]; Gastil et al., (1967)) interval from 31 to 2828 ppm. Ages for these

20 different crystals are calculated 200 times with different model parameters for $^{N17}D_0$, $^{N17}E_a$, $^{z}D_0$ and

$^{z}E_a$. To do this, we extracted every 100[th] model from the posterior ensemble of models generated

during the MCMC algorithm. This ensures that we are sampling model parameter space in proportion

to probability but also that the model correlations are reliably captured. Grain sizes are all set to 70 μm.

The results highlight the large spread in predicted ages for a single thermal history (Figure 4). The

overall spread in age is expected given the different temperature sensitivity of the crystals. However,

even for a specific amount of radiation damage there is still a large dispersion in the predicted ages. For

example, at [eU] values of about 1600 ppm, ages are expected to vary between 50 and 550 Ma. If a

range of grain sizes were also modelled for a specific [eU], the spread would be even larger.

Furthermore, a myriad of other factors will also contribute to this dispersion (see Fox et al., 2019 for a

discussion).

In order to highlight how uncertainty in diffusion parameters propagate to uncertainty in thermal

histories, we use QTQt (Gallagher 2012) and the data from McDannell et al. (2022). It is important to

note that our goal is not to determine a new thermal history from the data, but rather to assess

uncertainties related to the kinetics. For this reason, we only use only the ZHe data and do not

incorporate additional constraints. We use the 22 ZHe single grain age data for the sample "Minnesota"

of McDannell et al. (2022), and 2 sets of values for the 4 diffusion parameters based on the MCMC

sampling. These correspond to values of 129224.284308, 4.000427, 149756.087043, 4.130064 and

42388.297312, -5.000653 137904.535009, 3.253373 corresponding to $^{N17}E_a$, log10($^{N17}D_0$), $^{z}E_a$ and

log10($^{z}D_0$) for a high value for the frequency factor of N17 and a low value, respectively. We also use

the values originally proposed by Guenthner et al. (2013) as a reference. The priors for time and

temperature were specified to be 1500±1500 M.y. and 150±150°C respectively. We ran the sampler for

300k iterations burn-in and 300k post-burn-in, accepting models in the conventional MCMC way, such

that a proposed model of equal likelihood to the current model will be accepted irrespective of the

complexity. Results show that while the general trend of the cooling is very similar, the posterior

probabilities are all quite different (Figure 5). This has important implications for our ability to reliably

identify cooling signatures. In particular, the part of the thermal history that appears well resolved by

the data changes from 1000 Ma to 1500 Ma depending on the choice of radiation damage parameters.

This suggests that the resolution in estimating the timing of a given cooling event could be as large as

500 Ma, with obvious implications for resolving the timing of the Great Unconformity.



### 5. Implications

Our method to propagate ZRDAAM uncertainty highlights how variable the age-[eU] relationship might be for a given thermal history. In particular, our results suggest that the uncertainty of ZRDAAM-based thermal history inversions may be significantly underestimated. This has major implications for our ability to differentiate between subtle differences in temperature at specific times. In turn, it may be challenging to resolve cooling histories sufficiently to attribute the Great

Unconformity to Cryogenic Glaciations (McDannell et al., 2022) or geodynamic processes related to the break-up of Gondwana (Flowers et al., 2020).

The potential to underestimate age uncertainty for thermal modelling has been discussed by McDannell et al. (2022) and to some extent this can be accounted for in the inverse modelling software QTQt

(Gallagher, 2012). For example, if two dates have the same [eU] but their measured uncertainties do not overlap, QTQt can sample additional uncertainty for the measurements to account for this excess dispersion. However, if the two ages do not have the same [eU] concentration, the situation is more difficult. Either additional uncertainty can be assigned to the measurements by resampling a scaling factor ($> 1$) that multiplies the input errors. This tends to allow the predicted age-[eU] relationship to

pass through the observed data+resampled uncertainty. Or, alternatively, the thermal history can be adjusted to change the predicted age-[eU] relationship to try and ensure that the predictions fit the data, at least to within the error. The first option tends to produce simpler thermal histories than the second option, as the data fitting criterion is less strict. For example, McDannell et al. (2022)'s results for Pikes Peak highlight how models that ignore overdispersion appear to resolve a 700 Ma cooling

signature, which is smoothed out when the overdisperion is effectively reduced by adding excess uncertainty on some of the data.

We have shown the continuous spread of ages as a function of [eU] as a probability heat map for a specific history (Figure 4). In reality, most thermochronometric studies analyse 5-30 crystals for each

sample. We can illustrate the effect of model uncertainty by comparing two simulated datasets of 15 ages generated from the same thermal history. We can produce these datasets by sampling our age probability distribution randomly (Figure 6). Although these two datasets display overall similarities, there are subtle differences between their age-[eU] relationship over specific [eU] values. To accurately capture the spread in age for a single radiation damage value, many more thermochronometric samples



would need to be collected. To illustrate this point, we draw random samples from the probability distribution in figure 4, for [eU] values ranging from 1500 to 2000 ppm, and investigate the spread in age (Figure 6). The dispersion of the ages varies greatly with increasing sample size, converging to the predicted frequency distribution of figure 4. In our specific example, the distribution stabilises for sample sizes of 40 crystals. The need to accurately capture spread are especially important if ages need

to be averaged within [eU] bins to find acceptable paths as the uncertainty for the mean age is determined by the standard deviation (Flowers et al., 2020; Peak et al., 2021; Thurston et al., 2022). Ault et al. (2018) showed that simple visual identification under the microscope of the degree of metamictization is useful for obtaining good [eU] coverage and this approach could be adopted to ensure that multiple ages for the same [eU] are measured to get an idea of the spread in age.


The large uncertainties on the parameters controlling helium diffusion in zircon and the dramatic impact this has on temperature sensitivity highlights that this is important to consider. Currently, it is not practical to incorporate diffusion kinetic uncertainties in inverse models directly because this dramatically increases the volume of the parameter space that needs to be searched and would lead to

long run times. However, with the development of faster computers and parallelized inverse methods, this may be possible. Crucially, to ensure that we sample crystals with the same [eU] values to resolve diffusion kinetic parameters, many more ages per sample need to be analysed. For example, to accurately capture the spread in age for a relatively narrow [eU] range of 1500-2000 ppm, 40 crystals from this interval were required (Figure 6). A practical solution to avoid measuring so many crystals

per sample and running millions of simulations in an inversion is to use forward modelling. To do this, a single thermal history that is close to what might be expected for a specific area given prior knowledge could be used to assess expected age spread. This expected age spread could then be added to the age uncertainties used for inverse modelling. This procedure could be iterated to produce realistic uncertainties.


ZRDAAM has been calibrated using a limited number of diffusion experiments. Additional work is required to develop this dataset to capture diffusion kinetics at different radiation damage values. These experiments could also aim to replicate diffusion kinetics at previously measured radiation damage values to quantify the degree of dispersion. In addition, natural laboratories could be utilised to resolve

diffusion parameters: areas with known thermal histories can be exploited to predict ZHe ages by varying diffusion parameters; complementary thermochronometers can be leveraged to find thermal histories and diffusion parameters that match the observed data. Ultimately, by reducing the





uncertainties in helium diffusion kinetics using the constraints from man-made and natural laboratories, the timings of cooling events in the past can be resolved with more accuracy and precision.


**Code availability.** Codes required for this analysis can be requested from the authors.

**Data availability**. No new data were generated for this analysis.

**Author contributions.** MF designed the analysis, carried out the uncertainty modelling. AS, PV and AC contributed to the concepts discussed. KG carried out the QTQt modelling. All authors contributed

to writing.

**Competing interests**. The authors declare no competing interests.

**Acknowledgements**

We thank W. Guenthner for sharing data used for the calibration and for comments on an earlier

version of the manuscript. This study was supported by NERC (NE/N015479/1).

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



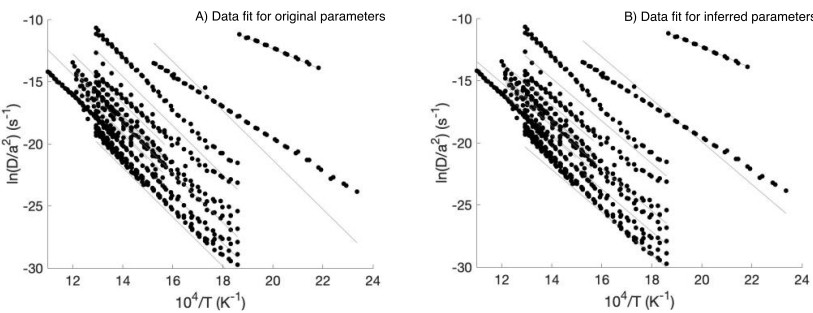


**Figure 1. Key parameters controlling how radiation damage controls diffusivity have been inferred from fitting Arrhenius relationships from step-degassing experiments. A) The fit to the step-degassing experiments for the model parameters inferred by Guenthner et al., (2013). B) The fit to the step-degassing experiments using model parameters extracted from the data using a Markov Chain Monte Carlo analysis. The model parameters are the minimum misfit model parameters and represent a single realization of the parameter set we infer.**






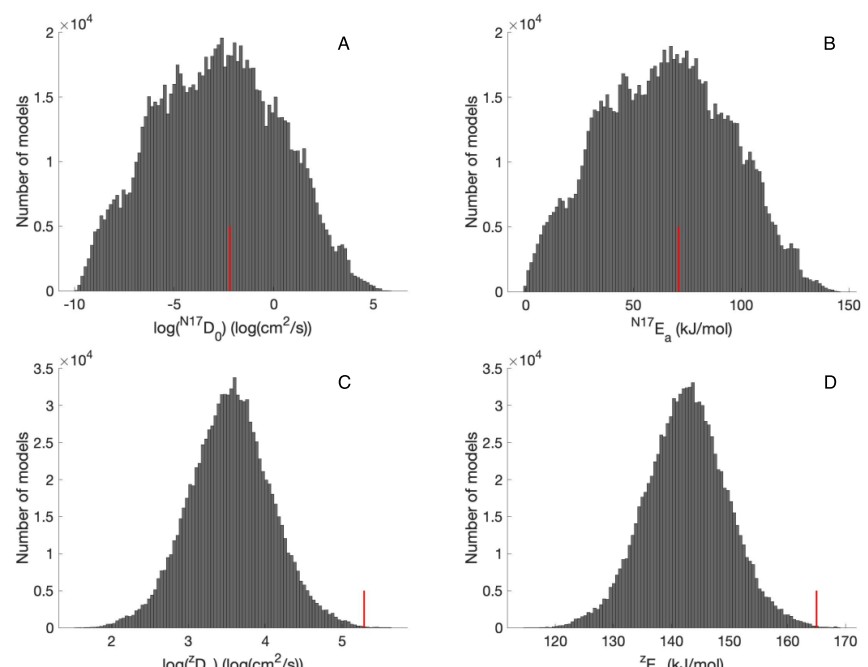

**Figure 2. The sampled marginal posterior distributions for the four diffusion parameters representing the two hypothetical crystals. A) and B) are the frequency factor ($^{N17}D_0$) and activation energy ($^{N17}E_a$), respectively, for a damaged, amorphous crystal. C) and D) are the frequency factor ($^{z}D_0$) and activation energy ($^{z}E_a$), respectively, for an undamaged crystal. The red lines show the values of these parameters used by Guenthner et al., (2013).**




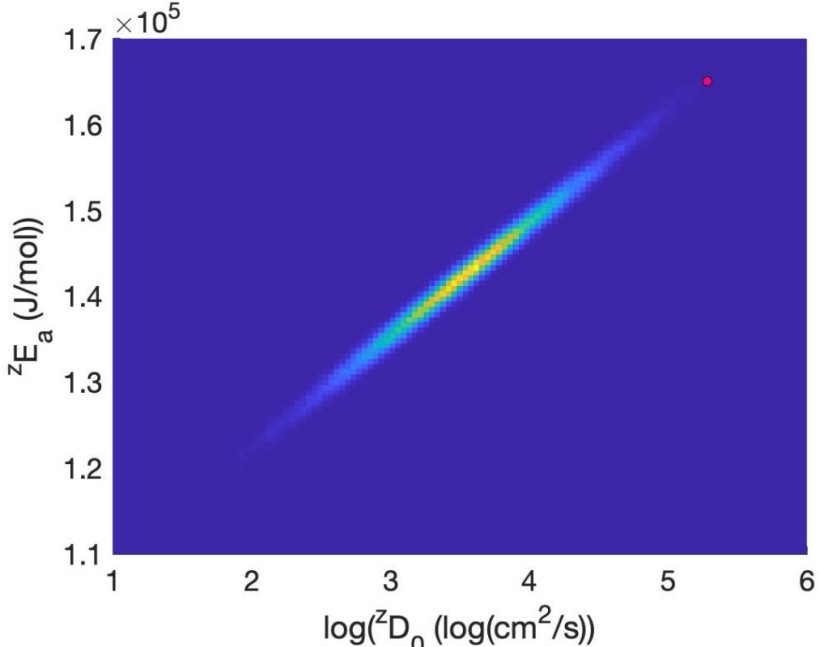

**Figure 3. Inferred model parameters from diffusion data are strongly correlated. Our approach to infer the diffusion kinetics of the hypothetical crystals using the radiation damage model maintains this correlation.**

**This is clearly illustrated with the diffusion parameters for the undamaged crystal. The pink spot shows the diffusion kinetics inferred by Guenthner et al., (2013) and shows that it is reasonably far from the center of our distribution, but still falls on the clear correlation trend we define. The colours are proportional to posterior probability with oranges reflecting highest probability.**





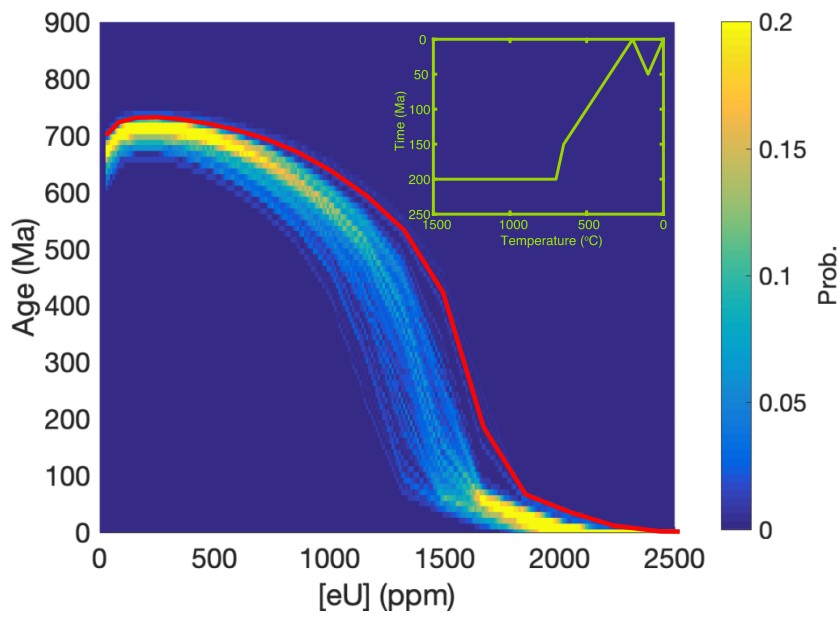


**Figure 4. Propagating uncertainties in the radiation damage model produces a wide range of ZHe ages for a specific amount of damage. The red line shows the predicted age-[eU] relationship using the canonical values of the radiation damage and annealing model. The continuous thermal history used to produce the**

**result is shown in the inset.**






## A) ZRDAAM Model Parameters

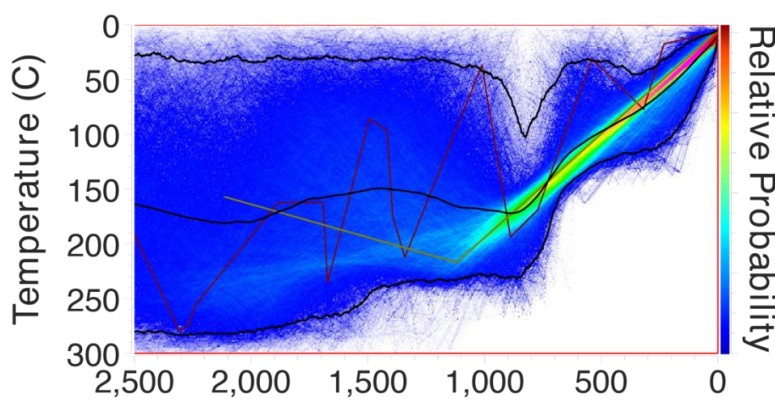

## B) High amorphous frequency factor

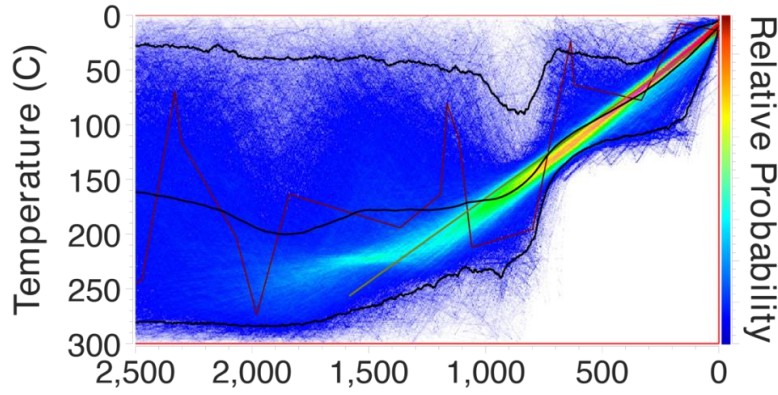

## C) Low amorphous frequency factor

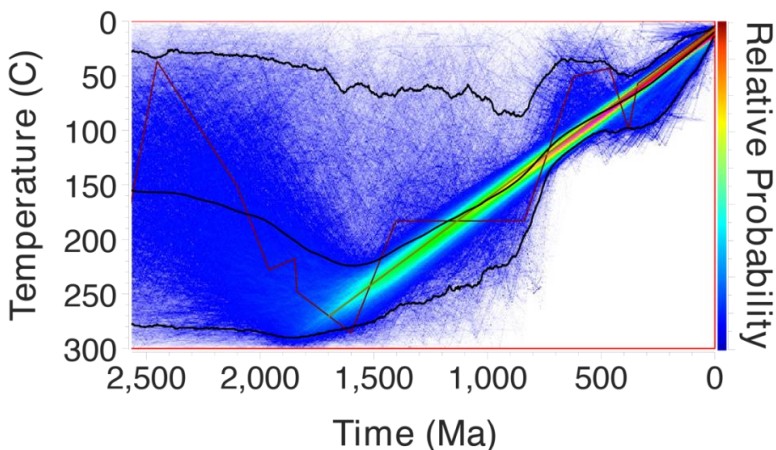





**Figure 5. Different parameter values for the damage model lead to differences in inferred thermal histories. A) The thermal history recovered from QTQt using the canonical values of the radiation damage model from Guenthner et al., (2013). The colours are proportional to posterior probability, the grey line is the maximum a posteriori model and the stepped line is the maximum likelihood model. The three curved lines are the expected model with the upper and lower credible intervals, please refer to Gallagher (2012) for**

**more details on QTQt. B) and C) The recovered thermal histories using different parameter sets with a high value and a low value of the amorphous frequency factor, respectively. The values for the two parameter sets are 129224.284308, 4.000427, 149756.087043, 4.130064 and 42388.297312, -5.000653 137904.535009, 3.253373 for $N17_{Ea}$, $\log10(N17_{D0})$, $ZEa$, and $\log10(Z_{D0})$. These exact values are drawn from the Markov Chain in order to ensure that they account for the complex model correlations. The overall patterns are**

**very similar, but the apparent resolution is different, resulting in different geological conclusions.**




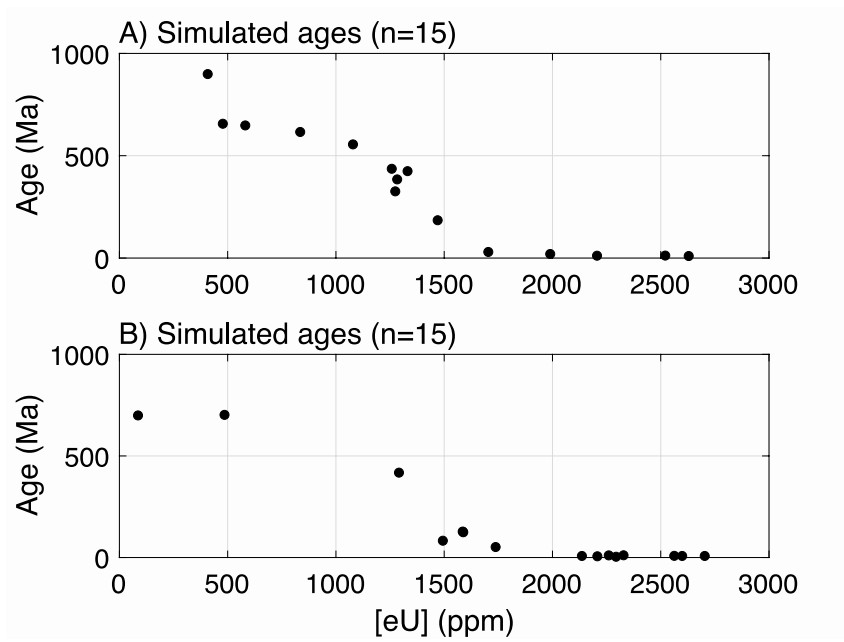

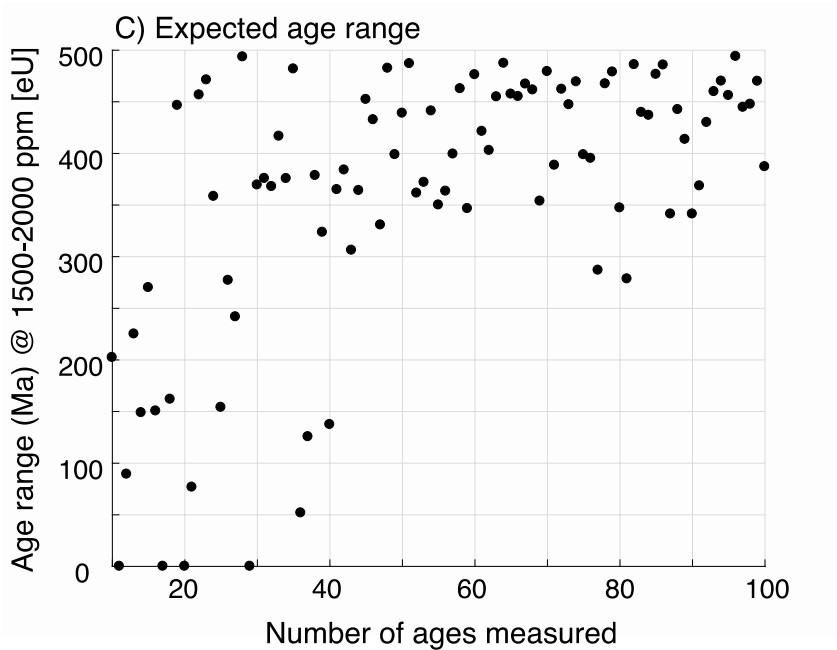

**Figure 6. Model realizations and expected ranges of ages. A & B) random samples are drawn from the probability distribution in Figure 4 to highlight the sorts of datasets that are expected given the typical number of ages measured on a single sample. The simulated ages are different between the two realizations**





**of a typical dataset. C) Many ages need to be sampled in order to accurately capture the spread in ages over**

**a specific [eU] bin. It is challenging to measure the spread because there is no easy way to estimate the**

**amount of radiation damage a crystal has accumulated until after the helium concentration has been**

**measured.**


