# Peer review of "Origin of Great Unconformity Obscured by Thermochronometric Uncertainty"

_Geochronology, 2022_

## Referee Comment (RC2)

**Review of Fox et al., "Origin of Great Unconformity Obscured by Thermochronometric Uncertainty"**

In this manuscript, Fox and co-authors present a detailed and useful discussion regarding the effects of uncertainties related to the Zircon Radiation Damage and Annealing Model (ZRDAAM) on thermal history models based on zircon (U-Th)/He data. They go on to suggest that these effects complicate thermochronologic models that have been used to constrain plausible causes for the Great Unconformity at the end of the Precambrian.

After a brief review of the reasons why some sort of ZRDAAM-like model is necessary to reasonably interpret zircon (U-Th)/He data, the authors correctly note that the original ZRDAAM formulation of Guenthner and co-workers (Guenthner et al., 2013), based on helium diffusion experiments on variably damaged zircons, has shortcomings that make it difficult to realistically propagate uncertainties in two experimentally derived helium diffusion parameters – the pre-exponential constant $D_o$ and activation energy $E_a$ – into ZRDAAM models and onward into thermal history models. In order to do this more quantitatively, they first present a revised ZRDAAM formulation in Section 3 of the manuscript. They show that diffusion parameters derived from experimental data using the new formulation mostly agrees with the diffusion parameters adopted by Guenthner et al. for a highly damaged ("amorphous") crystal, but differ markedly for an "undamaged" crystal (Figures 1 and 2). They attribute this to difficulties in estimating $D_o$ and $E_a$ independently using the approach of Guenthner et al. Most importantly, however the authors show that the uncertainties on derived parameters are quite large, much larger than the thermochronology community generally assumes.

Using the revised ZRDAAM formulation, Fox et al. then proceed to explore how such large uncertainties in radiation damage and annealing model parameters might affect thermal history models calculated using the QTQt software package of Gallagher et al. (Gallagher, 2012). Unfortunately, direct propagation of such uncertainties into inverse modeling of thermal histories using QTQt is computationally impractical (lines 302-305), but the authors present the results of efforts using simplified models showing that uncertainties of up to several hundreds of million years in the estimated timing of a cooling event should not be unexpected (!). They go on to point out, specifically, that these results have dramatic implications for studies such as those being conducted by several groups on the Great Unconformity.

I enjoyed reading this well-written contribution and feel it is an important step toward developing a robust appreciation of how confident we should be in the results of thermal history modeling studies based on datasets for which either zircon or apatite (U-Th)/He radiation damage modeling is required. I found no fault in the mathematics presented here and I trust the authors to have done the modeling carefully. I hope, however, that these findings are not used by non-specialists to conclude that thermal history modeling using thermochronologic data is practically useless because the uncertainties are so large. Fox et al. have tried a bit to guard against that, but they might be more explicit about that in the Implications section. The principal lessons I think we should learn are: 1) we should not overinterpret every wiggle in a modeled time-temperature path as having geologic significance; 2) we should not overestimate

the precision with which we know the timing of specific cooling (or reheating) events given the imprecision of derived diffusion and annealing parameters; and 3) we should all continue to work hard to improve our understanding of the impact of radiation damage on diffusion parameters through a combination of experimental *and empirical* studies. I was happy to see the nod toward also using natural laboratories to address these issues moving forward (lines 319-322). In fact, led by Alyssa Anderson, our research group at Arizona State performed such a study (Anderson et al., 2017), concluding that the Guenthner et al. formulation failed to predict ZHe closure behavior consistent with QTQt models of an independent multichronometer dataset for the McClure Mountain syenite of Colorado. However, Anderson and co-workers preferred to interepret this inconsistency as a consequence of variable and complex radiation damage zoning in individual McClure Mountain zircons rather than uncertainties in radiation damage. Certainly, however, uncertainties in the original ZRDAAM model might have played a role, as the Fox et al. work indicates. (It should be noted that, while the accuracy of the cooling history we derived has been disputed based on geological interpretations (Weisberg et al., 2018), the alternative proposed cooling history still remains inconsistent with the ZrnHe closure behavior predicted by the original ZRDAAM model (Anderson et al., 2018).) Our study and the current manuscript together remind us that the effects of radiation damage on ZHe closure behavior of ZHe is extremely complex and dependent on both the parameters we choose for modeling and the idiosyncracies of specific zircons. In such a climate of uncertainty, interpretive caution is advised.

*– Kip Hodges*

*References*

Anderson, A. J., Hodges, K. V., and van Soest, M. C., 2017, Empirical constraints on the effects of radiation damage on helium diffusion in zircon: Geochimica et Cosmochimica Acta, v. 218, p. 308-322.

Anderson, A. J., Hodges, K. V., and van Soest, M. C., 2018, Comment on 'Distinguishing slow cooling versus multiphase cooling and heating in zircon and apatite (U-Th)/He datasets: The case of the McClure Mountain syenite standard' by Weisberg, Metcalf, and Flowers: Chemical Geology, v. 498, p. 150-152.

Gallagher, K., 2012, Transdimensional inverse thermal history modeling for quantitative thermochronology: Journal of Geophysical Research, v. 117, p. 2156-2202.

Guenthner, W. R., Reiners, P. W., Ketcham, R. A., Nasdala, L., and Giester, G., 2013, Helium diffusion in natural zircon: Radiation damage, anisotropy, and the interpretation of zircon (U-Th)/He thermochronology: American Journal of Science, v. 313, no. 3, p. 145-198.

Weisberg, W. R., Metcalf, J. R., and Flowers, R. M., 2018, Distinguishing slow cooling versus multiphase cooling and heating in zircon and apatite (U-Th)/He datasets: the case of the McClure Mountain syenite standard: Chemical Geology, v. 485, p. 90-99.

---

## Referee Comment (RC3)

Review of Fox et al., Origin of Great Unconformity Obscured by Thermochronometric Uncertainty, *Geochronology*

This contribution is focused on better constraining the uncertainty associated with the zircon radiation damage accumulation and annealing kinetic model (ZRDAAM), and then evaluating how this uncertainty influences the ability to resolve thermal histories associated with development of the Great Unconformity. The manuscript begins by reviewing the existing ZRDAAM calibration, presents a new calibration based on the same kinetic dataset as used in the original ZRDAAM, uses a new approach to constrain ZRDAAM uncertainties, carries out inversion modeling of a dataset to explore the influence of ZRDAAM kinetic uncertainties on data interpretations, and concludes with some implications regarding the resolution of (U-Th)/He datasets in deep time.

This is a useful and informative contribution. I agree that there's opportunity to use the current Great Unconformity debate as motivation to improve kinetic model calibrations, which in turn can improve our ability to address problems like the origin of this feature.

This paper seems to approach the Great Unconformity problem from the direction of evaluating whether the thermochronologic data alone can resolve thermal histories in deep time. I completely agree that this absolutely is not possible without integration with other types of information, like that from geologic data.

I provide a variety of suggestions below that I think could be used to further strengthen the paper. Most importantly I highlight (described further below): 1) the need to more clearly and completely explain the new calibration, 2) that it's essential to include the Great Unconformity in any model that is supposed to explain the Great Unconformity, 3) that both the data being modeled and the predictions from the inversion models should be plotted, and 4) one aspect of the implications section that could be more valuable if expanded.

**Abstract**
Lines 13-14: "…about the origin of the Great Unconformity, a global erosional event that represents a period of almost a billion years at the end of the Precambrian."
         Suggest modifying this sentence. The Great Unconformity is a geologic feature, not an event. This sentence also asserts that it represents a global erosional event that represents a billion years – but the debate is actually about the timing and duration and whether it represents a global erosional event or not.

**Introduction**
Line 33: Keller et al. (2019) isn't appropriate to cite in this sentence because there are no thermochronologic data presented or interpreted in that contribution. Also suggest adding an e.g., before the refs, or adding more refs, because there are more papers that have used thermochronologic datasets to make interpretations about the Great Unconformity than those cited here.

Lines 35-36. "…in the Grand Canyon, the erosional event spans from circa 1200 to 250 million years…". This is the amount of missing time. The erosional event didn't span this entire interval. Also should use an e.g., for the reference, because many have noted this missing magnitude of time in the Grand Canyon.

Lines 38-40. "This approach highlighted that erosion rates increased across the North American craton during Neoproterozoic glaciation, supporting the hypothesis that…". It would be more appropriate to replace the word "highlighted" with "inferred". None of the datasets used in that paper require that erosion increased during the Snowball glaciation, as shown in the comment by Flowers et al. (2022). This is true even when one does not take into account kinetic model uncertainties, which I agree further increase the uncertainty in the inferred tT paths, and are the focus of this contribution.

Line 46: "…concepts of geochronology". Also involves concepts of diffusion. Suggest modifying to "…geochronology and diffusion."

Line 49: missing a word here, should be "diffusive loss".

Lines 43-51: Suggest emphasizing in this paragraph that the dates represent a time-integrated thermal history. This paragraph focuses only on simple exhumation scenarios, which differ from the deep-time studies discussed in this paper, that include multiple burial and erosion events.

Line 57: suggest rewording to "the transition from open-system to closed-system" given the phrasing of the previous sentences.

Line 59: Appropriate to cite Ketcham et al. (2013) here, along with Guenthner et al., 2013.

Line 67. "Both those in favour of a glacial (McDannell et al., 2022) and tectonic (Flowers et al., 2020) origin…"
        To be more accurate, suggest modifying to "Both those in favor of a globally synchronous glacial origin (McDannell et al., 2022) and regionally diachronous tectonic origins (Flowers et al., 2020)…" The text has framed the problem here and elsewhere as a debate over a single origin for this feature – but a critical aspect of the debate is whether or not there is indeed a singular origin or if there are multiple origins.

Lines 69-77. This is a great description, and also nicely highlights that annealing at higher temperatures after damage accumulation at low temperatures affects the retentivity and the date. It's this element that could be useful to bring into the geological framing in lines 43-51.

Line 79. Suggest changing RDAAM to ZRDAAM, as ZRDAAM was defined in previous paragraph. RDAAM refers to a widely used apatite radiation damage accumulation and annealing model, which isn't discussed in this paper.

Lines 81-85. "Parts of this history were reported to within less than 10 degrees between 700 and 250 Ma and then again from 15-7 Ma. It is unclear whether the data really provide such tight constraints on temperatures in the past…"
    - It's clear that the data don't provide such tight constraints on the temperatures in the past, because the model of their Figure 4 says that when the Great Unconformity developed that the basement was at 200 +/- 10 °C (so, 7-10 km deep in the crust, depending on assumptions), when we know unequivocally that the rocks were at the surface during deposition of the Cambrian sandstone on the basement to define the Great Unconformity. This model isn't set up to require that the Great Unconformity be part of the model. See Peak et al., 2022. This doesn't seem like a good example to use here.

Lines 87-88. "…known amounts of radiation damage." Suggest being more cautious here. Instead of saying "known amounts of damage", could say "…well-constrained amounts of radiation damage." For the Sri Lankan zircon, which I agree was a good sample to use in the Guenthner et al. study, the assumption of rapid cooling at 440-420 Ma to estimate radiation damage seems reasonable, but this is a long timescale with some uncertainty in the history. And even if one acquires Raman data, there is currently ambiguity in how to translate that data into the type(s) of damage in the crystal that matter for He diffusion.

Lines 94-95. "We show that natural variability in radiation damage annealing parameters causes ZHe ages to be dispersed even for crystals…" Delete "annealing"? As I understand it, this paper doesn't address variability and uncertainty in radiation damage annealing kinetics?

**The existing calibration of the radiation damage and annealing model**

Line 123. "…at a specific cooling rate." And assumed grain size.

Line 139. "..between the model parameters." Helpful to specify here that you mean zDo and zEa.

Lines 143-144. "However the accuracy of this model has only been assessed by looking at general trends in model predictions." Do you mean in Guenthner et al.? Or in the variety of studies that have aimed to explain data using this kinetic model? Provide some references here?

Lines 125-146. Clear explanation of the Guenthner model.

It would be worth explicitly noting somewhere that this contribution does not address uncertainties in the annealing model.

**A new calibration of the zircon radiation damage and annealing model**

This section is the crux of the paper and I think that more explanation is needed. I believe that I understand the Guenthner calibration, but I don't understand how the end member values for this new calibration (and thus their associated uncertainties) were obtained here based on the explanation provided. If what is presented here is indeed an improved approach as the authors are clearly arguing, then this is exciting and it should be explained with enough clarity so that those who generate the datasets used for model calibrations can understand and apply this method. If the authors would prefer not to lengthen the main text, then the appendix would be a great place for an extended description.

Lines 194-196. From the Figure 1A and 1B plots, it's not obvious that the revised calibration predicts the observed data better than the original calibration. It would be helpful to add some text here that explains what features of this plot that the reader should focus on to see this.

Figure 1. If I understand correctly, the purpose of this figure is to show how well the Arrhenius data for each sample used in the ZRDDAM calibration are fit with the original calibration and the ZRDAAM calibration of this paper. Right now, it's difficult to make this comparison across the two plots. This probably could be done more successfully by including a separate Arrhenius plot for each sample, labelling each Arrhenius plot with the sample name, and on the same plot including the diffusion kinetic data, the prediction from the original ZRDAAM calibration, and the prediction from the ZRDAAM calibration for that sample.

Lines 196-203 and Figure 2. Could you please explain more fully and clearly how these histograms were calculated. Again, this is the crux of the paper, since this calibration and the associated uncertainties are then applied to draw the main conclusions of this contribution.

The code used to do these calculations should be put in a supplement and made available for download so that others can reproduce these results and potentially apply it to other kinetic datasets.

**Propagating model uncertainties**

Lines 222-223. "Note, this implementation of the model has been used previously (Tripathy-Lang et al., 2015). It sounds like this model calibration was presented seven years ago in previous work? To what extent was it described there? This should be explained in the introduction.

Lines 224. "…an effective U concentration ([eU = [U] +0.24[Th]; Gastil et al., 1967))." I hadn't encountered this paper before, which interestingly uses eU to refer to "equivalent uranium" (although not effective uranium). That paper doesn't include an eU equation. Cooperdock et al. (2019) should be cited here for the equation, (eU = U + 0.238Th). Flowers et al., 2022 also lays out the equations for eU.

Line 241. "…and the data from McDannell et al. (2022)". Miltich (2005) should be cited for the data. McDannell et al. mined the Minnesota data from the Miltich (2005) undergraduate honors thesis without publishing them.

Line 242. "…for the sample "Minnesota"… These are actually multiple samples, not a single sample.

Lines 241-242 and Figure 5. "For this reason, we use only the ZHe data and do not incorporate additional constraints."
- This simulation is supposed to explain the Great Unconformity, but the highest probability time-temperature paths aren't at surface conditions when the Great Unconformity developed. In the final comparative statement that compares the three panels, the caption states "The overall patterns are very similar, but the apparent resolution is different, resulting in different geological conclusions." To me, the geological conclusion here should be that all of these model results are geologically meaningless because the highest probability tT paths violate the Great Unconformity, so all three models should be discarded. Could the authors either include the Great Unconformity in the models or better articulate why the Great Unconformity isn't honored in a model that is supposed to explain the Great Unconformity? This is now a repeated characteristic of many published QTQt models that are supposed to reproduce the Great Unconformity.

Figure 5. Please plot the data being modeled here and show how well the tT paths fit the observed data – for example, by making date-eU plots of observed vs. modeled data. See Gallagher (2016).

**Implications**

Lines 264-266. "In turn, it may be challenging to resolve cooling histories sufficiently to attribute the Grat Unconformity to Cryogenic Glaciations (McDannell et al. 2022) or geodynamic process related to the break-up of Gondwana (Flowers et al., 2020)." Yes, agreed that it's not currently possible to resolve cooling histories at this level with the "thermochronologic data alone" even when not considering uncertainties in kinetic models. This is why it is essential to integrate other types of information into models, such as geologic data. Flowers et al. (2020) didn't argue that we could resolve the cooling histories sufficiently without other information, as implied in this sentence. Perhaps you could modify to remove that implication?

Lines 268-281, and final sentence in this paragraph: "For example, McDannell et al. (2022)'s results for Pikes Peak highlight how models that ignore overdispersion appear to resolve a 700 Ma cooling signature, which is smoothed out when the overdispersion is effectively reduced by adding excess uncertainty on some of the data."
- There are interesting and important points made in this paragraph about how uncertainties can be accounted for in QTQt and the associated influence on the inferred tT paths. The final sentence that makes a vague reference to a figure in McDannell et al. (2022). It would be great if the authors would add a figure in this paper that helps to illustrate the valuable points

being made here rather than vaguely referring to a figure in that paper. Alternatively, this paragraph could be eliminated.

- The Pikes Peak models in McDannell et al. (2022) also violate the Great Unconformity relationship, as noted elsewhere about other published models.
- If this model is discussed, the I feel that it also is important to cite our 2022 comment on this paper. In that comment we show that entirely different tT paths, not captured in the McDannell et al. models, can explain the data.

Lines 283-314. This is a nice illustration of the dispersion expected in real datasets, and provides some of the rationale for the binning into synthetic grains as done by many who simulate (U-Th)/He data.

Code Availability. It would be appropriate to put the codes used for the calculations in this paper in a supplement and available for download so that others can reproduce these results and apply the approach to other kinetic datasets. This is now done so easily that it no longer seems appropriate to require an email to the authors to obtain the code.

I enjoyed reading this contribution and hope that the authors find these comments helpful to further strengthen the manuscript.

Becky Flowers, CU Boulder

References

Cooperdock, E.H.G., Ketcham, R.A., and Stockli, D.F., 2019, Resolving the effects of 2-D versus 3-D grain measurements on (U-Th)/ He age data and reproducibility: Geochronology, v. 1, p. 17–41, https://doi .org /10 .5194 /gchron-1-17-2019.

Flowers, R.M., Ketcham, R.A., Macdonald, F.A., Siddoway, C.S., Havranek, R.E., 2022, Existing thermochronologic data do not constrain Snowball glacial erosion below the Great Unconformities: *Proceedings of the National Academy of Sciences*, Letter to the Editor, v. 119, No. 38, https://doi.org/10.1073/pnas.2208451119.

Flowers, R.M., Zeitler, P.K., Danišík, M., Reiners, P.W., Gautheron, C., Ketcham, R.A., Metcalf, J.R., Stockli, D.F., Enkelmann, E., and Brown, R.W., 2022, (U-Th)/He chronology: Part 1. Data, uncertainty, and reporting: *Geological Society of America Bulletin* special volume on the *Reporting and Interpretation of Geochronologic data,* https://doi.org/10.1130/B36266.1.

Ketcham, R.A., Guenthner, W.R., and Reiners, P.W., 2013, Geometric analysis of radiation damage connectivity in zircon, and its implications for helium diffusion: The American Mineralogist, v. 98, p. 350–360, https://doi .org /10 .2138 /am .2013 .4249.

L. Miltich, 2005, Low temperature cooling history of Archean gneisses and Paleoproterozoic granites of southwestern Minnesota. Undergraduate thesis, Carleton College, Northfield, MN.

Peak, B.A., Flowers, R.M., Macdonald, F.A., and Cottle, J.M., 2022, Forum, Reply to Comment on: Zircon (U-Th)/He thermochronology reveals pre-Great Unconformity paleotopography in the Grand Canyon region: 50 (3): e544, https://doi.org/10.1130/G49965Y.1.

---

## Referee Comment (RC4)

**Systematic uncertainty and thermochronology of the Great Unconformity? A review of Fox et al. 2022, gchron-2022-23**

C. Brenhin Keller[1]

[1]Department of Earth Sciences, Dartmouth College, Hanover, NH 03755
* * *
The title of the manuscript by Fox and coauthors poses a clear question:

> *Can the origin of the Great Unconformity be resolved by thermochronology?*

A valid question! However, this is in detail actually quite a different question from that which this manuscript actually addresses, which might be more accurately stated as:

> *Can the origin of the Great Unconformity be resolved **by timing alone** using **single-chronometer ZHe thermochronology at a single location?***

While this may at first seem like a fine point, in the context of recent debate about the Great Unconformity it is absolutely critical that these two should *not* be conflated, given that the answers to these two questions are almost certainly *not the same*: specifically, the answer to the former question is (given enough data from different locations using different and/or multiple chronometers) quite likely **yes**, even though the answer to the latter question is quite likely **no**.

Even this more accurate latter question, however, might be somewhat misleading as a title, as it implies that the issues considered herein are somehow unique or particular to the question of the origin of the Great Unconformity. What Fox and coauthors do present is an analysis of the effects of diffusion model uncertainty on single-chronometer zircon helium time-temperature inversions, with the Great Unconformity chosen as a case study.

Moving past the title, this is a highly worthwhile undertaking. Systematic uncertainty is, in general, often overlooked and underappreciated in the geosciences – and diffusion model kinetics in thermochronology are no exception. More study of the implications of systematic uncertainties affecting thermochronometric t-T inversions (and more funding thereof!) is highly welcome. If recent heated debate regarding the Great Unconformity, and especially the possibility of thermochronologically resolving the timing of exhumation associated therewith [1, 2, 3, 4, 5] can help attract attention to such work, so much the better! Nonetheless, it is I think worth noting that this is far from the only application (or even the only high-profile one) to which questions of systematic thermochronometer uncertainty apply.

I do have some concerns regarding the representativeness of the presented t-T inversions (i.e., Figure 5); in particular, the strong diagonal path density which more or less entirely sidesteps the known Cambrian surface exposure suggests to me given my previous experience with this particular dataset that either the model has not fully converged on the stationary distribution (i.e., too short of a burn in burn-in), that a significant likelihood preference for simple paths has been applied (which it perhaps should not be in this case), or that data uncertainties have been overestimated (possibly due to running age uncertainty resampling on helium age uncertainties that have *already* been expanded by empirical resampling in McDannell et al. [3]), or some combination thereof.

Although it has long been common to run only some tens to at best low hundreds of thousands of steps of burnin, our recent experience with deep-time inversions, especially data rich ones, is that at the very

least 500k (e.g. [3]), but more preferably >1M (e.g., [5]) steps of burnin should be run for such inversions, along with at least 500k steps collected post-burnin. Some approaches such as simulated annealing (e.g., van Laarhoven and Aarts [6]) as tentatively implemented for zircon helium in Keller et al. [7] (the `Thermochron.jl` package also used in the Community Comment by McDannell [8]) may reduce the number of required steps, though this requires further study.

As the authors well know, assessing convergence is one of the hardest problems in Bayesian MCMC inversions, and it is always safer to err on the side of *more burnin*. This, along with ensuring that analytical uncertainties are not *doubly* expanded by sequential empirical and hierarchical resampling is my biggest requested methodological change to the current manuscript. These issues are in my view entirely addressable and would not detract from the present manuscript.

One *might* propose that in assessing the influence of kinetic model uncertainty on inversion results, it in some sense may not matter whether the distribution has become fully stationary or whether it reproduces unambiguously known geologic constraints; all that maters is that the inversion results are different with different kinetics. However, if this exercise is going to be broadly useful, what we all are really going to be most interested in is how it applies to inversions that *are* fully stationary, and geologically valid. In other words, the current models with completely overshoot the uncontroversially established Cambrian surface exposure are not representative of how a Bayesian t-T inversion would or should be used in practice for these data, and thus should not form our basis for assessing the importance of systematic kinetic uncertainties on such inversions. Indeed, while I suspect fully burned-in results for these data will be at least probabilistically consistent with the known Cambrian exposure history, it would be most valuable to consider (as [3] did) models that both do and do not impose a Cambrian exposure constraint.

That is to summarize, on the technical side I would recommend that the authors:

- Ensure that uncertainties have not been redundandly expanded

- Run all models with at least 500k steps burn-in (preferably ~1M)

- For each set of diffusion parameters, run both an "unconstrained" model as well as one that imposes a Cambrian exposure constraint (say, 0-40C, given Cambrian surface temperature uncertainties)

Beyond these entirely tractable issues, there is one more major concern I have at present: the current framing of the manuscript *appears* to imply (possibly unintentionally?) that McDannell et al. [3] were unaware of systematic uncertainties in the zircon helium system, or that these issues significantly undermine or invalidate the results of McDannell et al. [3]. This I find somewhat odd given that a key point of McDannell et al. [3], as opposed to some previous studies, is that a single-chronometer inversions from single localities in isolation are likely insufficient to resolve the origin of the Great Unconformity, and that instead *multichronometer inversions* (which reduce the impact of systematic uncertainty in diffusion kinetics since different chronometers have different systematic kinetic uncertainties, and also more generally increase t-T resolution by widening the range of effective diffusivities and annealing rates considered) and particularly *the spatial pattern of exhumation* are key to obtaining meaningful insight into the origin of the Great Unconformity via thermochronology. This first clearly discussed on Page 2 of McDannell et al. [3], where we note:

> *First, the uncertainty of time-temperature (t-T) paths derived from a single thermochronometer can be large for older rocks a problem sometimes exacerbated by the use of suboptimal inversion methodologies – making it difficult to discern between glacial and tectonic drivers by timing alone. Second, the magnitudes of both glacial and tectonic erosion are expected to be spatially heterogeneous. Fortunately, however, glacial and tectonic processes predict distinct spatial patterns of exhumation with tectonic erosion focusing in tectonically active regions near cratonic margins and ice-sheet glacial erosion focusing in regions of wet-based icenamely, in the models of Donnadieu et al. (33), broad regions of the low-latitude cratonic interiors away from ice divides, narrowing to a more "hit-or-miss" pattern at cratonic margins where basal slip is focused into only a few rapid outlet ice streams, as is observed*

> *at modern Greenland and Antarctic ice margins. Consequently, to resolve the relative contributions of all such climatic and tectonic drivers of erosion in the Neoproterozoic, not to mention their potential interactions, we require higher-resolution tT paths from localities that can address the spatial pattern of Neoproterozoic exhumation at a global scale. [ … ] The use of multiple thermochronometers with varying temperature sensitivities is critical for such deep-time applications.*

and again on Page 6, where we further note that it is indeed not only thermochronometric uncertainty, but also uncertainty in the timing of tectonic forcings that makes a spatially-aware approach all the more critical:

> ### Spatial Patterns of Tectonic and Glacial Erosion of Continents
>
> *McDannell et al. (55) and DeLucia et al. (36) came to the conclusion that kilometer-scale Neoproterozoic exhumation occurred after 1 Ga within the North American interior and linked this to formation of the Great Unconformity due to Rodinian geodynamics and/or snowball Earth glaciations. These two hypotheses are not mutually exclusive – it is possible that both tectonics and glaciation contributed to global Earth system disruption (80, 81) during formation of the Great Unconformity. Glaciation would be most effective as a driver of erosion in regions with preexisting topography (be it from rifting or orogeny); therefore, erosional synergy between tectonics and ice sheets is a possibility (82). […] Direct and meaningful comparisons between tectonic and glacial unconformity hypotheses are complicated by the fact that there are precise estimates for the timing of Snowball glaciations (23), whereas the timing and duration of Rodinia assembly and breakup remain incompletely understood due to discrepancies between paleomagnetic and geologic data (11, 83, 84). Rodinia assembly and breakup occurred episodically and diachronously over at least 250 Ma for each phase, with timing dependent upon location (10, 11).*

and again on Page 7:

> *Cratonic interiors provide the only location to truly test and differentiate the hypotheses of pre-, syn-, or post-Cryogenian formation of the Great Unconformity. Timing is a key component of this signal, but spatial pattern and magnitude of exhumational rock cooling are also critical. Tectonic rifting and glacial erosion will produce opposing spatial patterns of exhumation and different magnitudes of crustal unroofing across a continent. The majority of exhumation associated with supercontinent assembly and breakup would be limited to compressional orogenic belts and extensional (faulted) rift margins, respectively. Rifting will show large exhumation narrowly restricted to continental margins, where tectonic activity is highest, whereas stable continental interiors will experience little to no erosion or even deposition.*

Other key factors noted by McDannell et al. [3] in differentiating between tectonic and glacial causes include the apparent absence of an equivalent "Great Unconformity" phenomenon associated with the breakup of Pangea (or perhaps relatedly, how exactly it is that the "continental rifting" part of the supercontinent cycle is supposed to cause more erosion and exhumation than the "continental collision" part).

Subsequently, we continued to reiterate the importance of relying on more than timing alone in our more recent contribution on the subject [5], wherein we note:

> *The Canadian Shield margin displays many features indicative of late Neoproterozoic rift-related tectonism and is, in principle, consistent with a mantle plume model (i.e., Sturrock et al., 2021), including pre-rift doming, pervasive faulting, dike emplacement, and syn/post rift deposition (Cawood et al., 2001; McClellan and Gazel, 2014). Evidence for such events is, however, absent within the stable cratonic interior. We maintain that tectonic phenomena such as rifting or distal plume impingement (Sturrock et al., 2021) are unlikely to drive >36 km of exhumation within the continental interior, which is far from the western Laurentian*

> *Cordillera margin and more than 2000 km inboard of the Iapetan rift margin. The inferred magnitude of erosion is also greater than models of dynamic topography commonly predict (<3 km; Braun et al., 2013).*
>
> *While these results appear to support a broad pattern of denudation across disparate, stable cratonic regions of North America (e.g., McDannell et al., 2022), we by no means rule out variability in the timing and magnitude of cooling; indeed, such variation is expected even in a glacial endmember hypothesis.*

The only case in which we expect a single t-T inversion could significantly inform our understanding of the Great Unconformity would be in the case of an undisputedly stable cratonic interior where no tectonic forcings for kilometric exhumation are plausible regardless of some nontrivial uncertainty in timing – but even then a multichronometer inversion as in [5] is far preferable, and even then is most meaningful only in the context of a broad spatial pattern supported by numerous independent inversions at different cratonic and marginal localities.

That is, to summarize key points from McDannell et al. [3] and [5] as quoted above:

- We expect that single-thermochronometer inversions have inadequate timetemperature resolution to differentiate between geologic and tectonic causes of exhumation by timing alone, and do not rely upon them (indeed, only one out of our seven presented t-T models between these two papers relies on a single chronometer alone, in significant contrast to the work of some others on the same subject).

- Multichronometer inversions are better, but still technically inadequate to resolve the debate in isolation / by timing alone, considering that tectonic and glacial exhumation may temporally *coincide.*

- Instead, the *spatial pattern of exhumation between tectonically active cratonic margins and tectonically stable cratonic interiors* is most key to differentiating between glacial and tectonic exhumation.

Consequently, I fully agree that systematic thermochrometer uncertainty is an important consideration, but have already taken major steps to ensure that it does influence our conclusions, something which the current manuscript *appears* to suggest the opposite of.

Finally I have some concern from a statistical perspective with the robustness and geologic significance of the "onset of cooling" metric discussed in the current manuscript; the time of half-cooling or width of the distribution at half-cooling is likely more useful, robust, and geologically meaningful in practice for reasons discussed in more depth in McDannell [8]. These issues do not mean that the path density prior to a cooling event cannot be used when discussing variation in the shape of a t-T inversion in the abstract, but should (I would propose) be considered and perhaps acknowledged whenever there is a clear geologic context being discussed.

Given the broader implications of uncertainties in diffusion model kinetics far beyond the Great Unconformity, an additional demonstration of the effects of varying diffusion kinetics on another dataset of interest might also be welcome, though I would not insist upon it. Overall, I find that the current manuscript addresses an important and often little-discussed factor that is quite relevant to the field of thermochronology as whole, despite some tractable potential issues in t-T inversion, framing, and titling.

I recommend Revisions and would be happy to consider a revised manuscript.

C. Brenhin Keller

**References**

[1] C. Brenhin Keller, Jon M. Husson, Ross N. Mitchell, William F. Bottke, Thomas M. Gernon, Patrick Boehnke, Elizabeth A. Bell, Nicholas L. Swanson-Hysell, and Shanan E. Peters. Neoproterozoic glacial origin of the Great Unconformity. 116(4):1136–1145, 2019. doi: 10.1073/pnas.1804350116.

[2] Rebecca M. Flowers, Francis A. Macdonald, Christine S. Siddoway, and Rachel Havranek. Diachronous development of Great Unconformities before Neoproterozoic Snowball Earth. 570:201913131, 2020. doi: 10.1073/pnas.1913131117.

[3] Kalin T. McDannell, C. Brenhin Keller, William R. Guenthner, Peter K. Zeitler, and David L. Shuster. Thermochronologic constraints on the origin of the Great Unconformity. 119(5):e2118682119, 2022. doi: 10.1073/pnas.2118682119.

[4] C. P. Sturrock, Rebecca M. Flowers, and Francis A. Macdonald. The Late Great Unconformity of the Central Canadian Shield. 22(6):e2020GC009567, 2021. doi: 10.1029/2020GC009567.

[5] Kalin T. McDannell and C. Brenhin Keller. Cryogenian glacial erosion of the central Canadian Shield: The late Great Unconformity on thin ice. pages 1–23, 2022. doi: 10.1130/G50315.1.

[6] Peter J. M. van Laarhoven and Emile H. L. Aarts. Simulated annealing. In Peter J. M. van Laarhoven and Emile H. L. Aarts, editors, *Simulated Annealing: Theory and Applications*, Mathematics and Its Applications, pages 7–15. Springer Netherlands, 1987. ISBN 978-94-015-7744-1. doi: 10.1007/978-94-015-7744-1_2.

[7] C. Brenhin Keller, Kalin T. McDannell, William Guenthner, and David L. Shuster. Thermochron.jl: Open-source time-Temperature inversion of thermochronometric data. 2022. doi: 10.17605/OSF.IO/WQ2U5. URL https://osf.io/wq2u5/.

[8] Kalin T McDannell. A comment on Fox et al. gchron-2022-23: Is zircon (U-Th)/He kinetic model uncertainty [only] an issue for thermochronometric resolution of the Great Unconformity? pages 1–11, 2022. doi: 10.5194/gchron-2022-23-CC1.

---

## Community Comment (CC1)

**A comment on Fox et al. gchron-2022-23:**
**Is zircon (U–Th)/He kinetic model uncertainty [only] an issue for thermochronometric resolution of the Great Unconformity?**

**Kalin T. McDannell**

Department of Earth Sciences, Dartmouth College, Hanover, NH USA

kalin.t.mcdannell@dartmouth.edu

The manuscript by Fox et al. highlights the impact of uncertainty on time-temperature ($t$–$T$) inversions with respect to the widely used (U–Th)/He kinetic model that describes radiation damage effects on $^4$He diffusion in zircon (ZRDAAM; Guenthner et al., 2013). The focus on diffusion kinetic uncertainty is timely and commendable—hopefully stimulating further work to understand foundational aspects of zircon thermochronometry. A similar conclusion has been discussed for nearly a decade in the published literature regarding kinetic model calibration and uncertainty (e.g., Powell et al., 2016; Anderson et al., 2017; Johnson et al., 2017; Mackintosh et al., 2017; McDannell et al., 2019; Guenthner, 2021), but to date, few attempts have been made to formally account for uncertainties directly in the most commonly used thermal history modeling programs, nor have many more comprehensive laboratory diffusion experiments been undertaken to better understand how radiation damage affects diffusivity for a broader suite of natural zircons (e.g., Ginster et al., 2019). Kinetic uncertainties extend to the apatite (U–Th)/He system as well (e.g., Flowers et al., 2009; Gautheron et al., 2009; Fox and Shuster, 2014; Recanati et al., 2017; Willett et al., 2017; Guo et al., 2021). Thus, (U–Th)/He kinetic uncertainty has been a well-known problem that perhaps has not been addressed more decisively because the models have been considered good enough for most geologic applications.

**1 Kinetic model uncertainty**

Estimates of the effects of kinetic uncertainty for the (U–Th)/He system are rarely, if ever performed. Empirically derived results suggest $\alpha$-damage kinetics can be explained to first order by general characteristics of fission-track annealing (Guenthner et al., 2013, Ketcham et al., 2013), but there are differences in detail and gaps in our understanding remain (Ginster et al., 2019; Guenthner, 2021). Thus the ZRDAAM kinetic model recalibration presented in the Fox et al. preprint is also by definition imperfect. So it becomes a question of what level of kinetic model uncertainty are we willing to live with and when does it cause significant $t$–$T$ inversion inaccuracy? This is probably a challenge relevant to all timescales but it may be more important for certain geologic scenarios (e.g., Guenthner, 2021) and becomes especially difficult to quantify in deep time—since geological benchmarks are scarce and laboratory kinetic extrapolations become murky (Ketcham, 2019). Regardless of personal bias, seemingly casting "thermochronometric uncertainty" as only a concern for resolving the origin of the Great Unconformity is inappropriately narrow*.

**2 Parameter correlations**

The authors raise important points about parameter correlations and how different kinetics may change model results due to differences in damage annealing. Correlations between ZRDAAM diffusion kinetic parameters such as activation energy ($E_a$) and frequency factor ($D_0$) are important for assessing model accuracy and addressing uncertainties—yet extrapolations between theoretical minimally damaged and highly damaged amorphous zircons are still based on real, but limited, laboratory data. So while it is a useful exercise, it is nonetheless inhibited by the data grounding the established radiation damage relationship. Of course, $E_a$ and $D_0$ are dependent on time-temperature conditions. Time and temperature are also correlated and any change in temperature at one time can be compensated
* * *
*A detail worth noting is that Fox et al. stated (lines 91–92): "...*the uncertainties in the radiation damage model make it challenging to accurately infer the timing and magnitude of unconformities in the deep past*". Thermochronological methods lack the temperature sensitivity to determine the final erosional event that results in an unconformity, and the erosional surface itself is inherently a feature terminated and preserved by sedimentation. McDannell et al. (2022a) were not 'dating the unconformity' but were instead placing limits on the timing, magnitude, and most importantly the spatial pattern of widespread rock cooling and exhumation that led to formation of the Great Unconformity in North America.

by an opposing temperature change at another time (e.g., Willett, 1997). Isolating kinetic uncertainty in models is important, however, ignoring other factors that may affect $t$–$T$ inversions makes it hard to evaluate the absolute effects and practicality of such measures.

- Does reduction of the uncertainty of the diffusion data for the critically important damaged N17 zircon (lines 189–192) drive the 'excellent' MCMC recovery of $E_a$ and $D_0$ (their fig. 2), and does this in any way have an effect on the poorer recovery (with respect to the canonical values) of the parameters for the undamaged crystal?
- Can the Fox et al. ZRDAAM $E_a$ and $D_0$ calibration values simply be resampled using MCMC in a Bayesian $t$–$T$ inversion, and if so, how would that affect thermal history recovery?

**3  The effects of additional constraints on inversions results?**

Fox et al. presented QTQt inversions (Gallagher, 2012) without imposed constraints (latter commonly represented as $t$–$T$ boxes through which candidate histories must pass). To be clear, the so-called "unconstrained" models in McDannell et al. (2022a) were explicitly shown to assess the $t$–$T$ sensitivity of the data during recursive modelling and comparison of alternate models with $t$–$T$ boxes. Gallagher (2021) most recently discussed this QTQt modelling strategy in some detail. Other commentary has improperly dismissed that class of models as 'invalid' due to misunderstandings about their meaning and purpose (Flowers et al., 2022). However, such exercises prove useful and informative for validation of near-endmember models against the geologic record, rather than just simply forcing the data to conform to an uncertain, presupposed geologic model (see McDannell et al., 2022b). Any models presented in the former way should be viewed within the bounds of the kinetic assumptions (that are held fixed between inversions). Evaluating different inversion parameterizations with other geological constraints and/or associated uncertainties are valuable and motivationally transparent. Unconstrained inversions without many $t$–$T$ boxes are undoubtedly affected by uncertainty and parameter correlations (as are those with boxes). Yet, systematic uncertainties may pose a more dubious problem for inversions that enforce *many* optimistically certain $t$–$T$ constraint boxes based on interpretive assumptions.

- How do known physical geologic constraints affect the Fox et al. QTQt inversions? It is understood that examining kinetic parameter uncertainty in isolation was a goal of the paper, however, the effects of that uncertainty on the thermal histories would likely change with imposed constraints—this should be investigated and compared to the baseline case.
- Do geologic constraints imposed in a thermal history inversion impact the covariance/correlations between (kinetic) parameters?
- How do the observed vs. predicted zircon (U–Th)/He (ZHe) dates compare for the different QTQt inversions (i.e., Gallagher, 2016)?
- Does inverting multiple thermochronometers (as was done in the McDannell et al. 2022a MRVT models) reduce model non-uniqueness and change $t$–$T$ resolution?

**4  Posterior probabilities**

It is unclear how much the Fox et al. thermal histories actually change in detail (their fig. 5), and if such changes quantitatively impact interpretations for parts of $t$–$T$ space where the thermochronometer data are most sensitive. For example, their QTQt models mostly show differences in the pre-1000 Ma thermal history for the Minnesota River Valley Terranes (MRVT; McDannell et al., 2022a), which is not well constrained by data due to Neoproterozoic (and later) thermal resetting. However, by our account, the late Neoproterozoic cooling and episodic Phanerozoic reheating history is quite similar between the models implementing the original ZRDAAM kinetics and recalibrated high/low amorphous frequency factor kinetics. Fox et al. asserted that (line 251): "*Results show that while the general trend of the cooling is very similar, the posterior probabilities are all quite different (figure 5).*" The relative posterior probabilities seem comparable across all models except for the extremely linear regions of high probability (see below). The first-order differences in the recovered history styles, or at least the posterior probabilities of the accepted paths compared to the inversions in McDannell et al. (2022a) [that were run for much longer in QTQt], indicate that there may be procedural reasons for these discrepancies.

- Could the posterior probability change with increased MCMC sampling?

- Could the number of model iterations required for the posterior distribution to become stationary change depending on the kinetics?

**4.1 MCMC burn-in and ZHe uncertainties**

In all three of the Fox et al. QTQt models, the regions of $t$–$T$ space with the highest relative probabilities are very "linear"—a possible cause for this could be that the burn-in for the inversions was too short (and/or restarted QTQt inversions began with a poor model). We discovered analogous posterior probability behavior related to burn-in in preliminary QTQt tests for the MRVT samples (McDannell et al. 2022a; published models $\geq$ 500,000 burn-in/post-burn-in iterations). We ran QTQt model tests with up to 1,500,000 burn-in iterations and 1,000,000 post burn-in iterations—the accepted paths typically gained structure with longer run times. Much longer burn-in periods are required for deep-time inversions spanning billions of years with large (multi-)thermochronometer datasets.

Another likely reason for the linearity in the regions of high relative posterior probability in the accepted $t$–$T$ paths may be that, as far as we can tell, the Fox et al. QTQt models allowed the ZHe age uncertainties to be resampled up to 1–2$\times$ the input age error (K. Gallagher, pers. comm.). Fox et al. used the same the same Minnesota QTQt input ZHe data file was used in McDannell et al. (2022a). The problem with that approach is that the Minnesota dates in McDannell et al. (2022a) already underwent a form of Empirical Bayes error resampling prior to inverse modelling (see fig. S15 in that paper). The internal analytical uncertainties were used to calculate the 'external uncertainty' by creating a Gaussian kernel or normal probability density function in eU space centered on each uncorrected ZHe date (eU = effective uranium = U+0.238*Th). A 100-ppm eU kernel was taken to represent the range over which zircon grains with similar eU should have similar ages and the Empirical uncertainty was estimated by summing the internal and external uncertainties in quadrature (code available: `https://github.com/OpenThermochronology/EmpiricalBayes`).

Therefore, Fox et al. allowed the uncertainties on the input data to be much too large, which essentially causes a loss of apparent complexity in the observed date-eU pattern and allows the data to be easily reproduced—resulting in more simple $t$–$T$ paths being accepted (regions of $t$–$T$ space with high relative probability are linear). For example, the oldest 770 Ma zircon had an input Empirical Bayes uncertainty of $\pm$ 100 Myr but this was allowed to be sampled up to $\pm$ 200 Myr in the Fox et al. inversions. Due to this, we assume that the fits between the observed and predicted data are poor (i.e., many of the predictions are at the margin of acceptability), but that is unable to be evaluated in the preprint. A similar discussion in McDannell and Keller (2022) touched on the issue of uncertainty estimation for apatite (U–Th)/He data and how if uncertainties become too large then all $t$–$T$ sensitivity is lost (see their Fig. S1). In QTQt this can result in simple linear histories being accepted more often. Such a model may be interpreted as geologically meaningful but that may not be appropriate (e.g., Gallagher, 2021). Simple $t$–$T$ models are merely due to inadequate sensitivity/resolution, which could be sourced from the kinetics or the chronometer data. These outcomes are not just limited to Bayesian methods. Other software that utilize pure Monte Carlo search methods instead rely on many "exploration boxes" to delineate the model space, therefore, loss of sensitivity due to outsized data errors would probably never be recognized by the modeller. In many cases this would also be a welcome effect because it would allow more paths to be found more easily with a nondirected MC algorithm—this means that boxes (based on assumptions) could have more influence on the thermal history results than the (U–Th)/He data.

**4.2 Timing of cooling**

Fox et al. expressed (lines 96–98): "*Using QTQt, we show that different diffusion kinetics can lead to the onset of cooling for resolved thermal histories from inverse methods varying by hundreds of millions of years.*" and lines (253–254): "*In particular, the part of the thermal history that appears well resolved by the data changes from 1000 Ma to 1500 Ma depending on the choice of radiation damage parameters.*" These statements seem based on interpretation of where the initial timing of higher relative probability begins within the different QTQt models. Considering the potential limiting circumstances surrounding their date uncertainties being too large (and/or incomplete burn-in?) this seems a tenuous conclusion. Their interpretation overlooks the consistent Neoproterozoic cooling present in all of their models. In addition, there is no obvious reason why 'cooling onset' would necessarily correlate with high posterior probability. The time of peak cooling when the first derivative of a cooling curve is maximized is perhaps a better metric to consider (fig. 1). The apparent differences in resolution in their models is not necessarily because of the choice of radiation damage parameters. The fact that Neoproterozoic cooling is present in the Fox et al. models

but low probability suggests that there are possible issues surrounding burn-in or overall poor $t$–$T$ resolution due to overestimated ZHe uncertainties.

[Figure]

**Figure 1:** Schematic time-temperature plot showing three cooling curves of decreasing cooling rate from high to low temperature. In thermochronological inversions typically comprising a group of possible solutions, variable rate is a source of uncertainty and cooling onset is mostly controlled by data sensitivity (i.e., how well the model $t$–$T$ path resolves some true cooling signal). For example, if the true total cooling magnitude is 200°C, and the data are only sensitive to temperatures $\leq$ 100°C, then the apparent cooling onset with be temporally biased by the difference in time between the true cooling onset and data sensitivity onset, proportional to the slope of the cooling curve. Thus in this scenario, cooling onset would only be accurate for near instantaneous cooling and would be highly inaccurate for situations involving slow cooling—more similar to that expected in a cratonic setting like Minnesota. Moreover, if chronometer data are low sensitivity then direct bias can be introduced by $t$–$T$ constraint box arrangement portraying a seemingly well-resolved but inaccurate onset of cooling. Refer to McDannell and Keller (2022) for further discussion. It is our opinion that cooling onset is difficult to interpret and it is arguably less important than the time of 'peak cooling' for an overall cooling signature—when the first derivative of the cooling curve is maximized. The time of peak cooling is the same for all cooling curves shown here, as is the time of the true cooling event, yet with dramatically different times of "cooling onset".

**4.3 Continuous spread of ages as a function of eU**

Fox et al. also discussed an aspect of the measured ZHe data that directly plays into thermal history recovery—the impact of the spread in zircon eU—and if the spread is narrow, sensitivity is limited and modelled histories are more uncertain (their fig. 6). This was also reviewed broadly in McDannell and Keller (2022); see their supplementary material. Fox et al. stated "***Many ages need to be sampled*** *in order to accurately capture the spread in ages over a specific [eU] bin*" and also said (line 295): "*The need to accurately capture spread are especially important if ages need to be averaged within [eU] bins to find acceptable paths as the uncertainty for the mean age is determined by the standard deviation.*" We would argue that their findings actually make a clear and obvious argument against binning ages by eU and averaging them in the first place. The need to capture the spread in ages supports collection of more ZHe data, not less by arbitrary means. The practice of eU binning and averaging is increasingly common but it is an ad hoc attempt to circumvent other statistical limitations (see McDannell et al., 2022b). Therefore, if eU binning is performed, the assertion that thermochronometer data are inherently "low resolution" is an oversimplification and without merit—since a natural outcome of averaging is information loss. All thermochronometers have fundamental limits on $t$–$T$ resolution, which is the primary reason to apply *multiple* thermochronometers in deep time (McDannell et al., 2019; McDannell and Flowers, 2020).

[Figure]

**(a)** Canonical ZRDAAM Minnesota inversion without geologic constraints. **250,000** burn-in iterations and 250,000 post-burn-in iterations.

[Figure]

**(b)** Canonical ZRDAAM Minnesota inversion without geologic constraints. **350,000** burn-in iterations and 250,000 post-burn-in iterations.

[Figure]

**(c)** Canonical ZRDAAM Minnesota inversion with near-surface constraint at 560 ± 80 Ma between 0–50°C (black dashed box).

**Figure 2:** Inversion results for the Minnesota zircon (U–Th)/He data using `Thermochron.jl` and the canonical ZRDAAM kinetics. Cryogenian cooling is a consistent signal in all models. This is less clear in panel (a) due to incomplete burn-in; note the similarity with the Fox et al. QTQt models. Relative probability is proportional to path density, where warmer colors and higher saturation indicate more thermal histories pass through that region of $t$–$T$ space (i.e., higher marginal posterior probability). The color scale is the normalized path density (minimum value of 0 is equal to no paths, and a maximum value of 1 is equal to the upper 95th percentile of path density). Except for panel (b), the Markov chain was run for 500,000 total iterations with a burn-in of 250,000 iterations. The prior was 400–0°C and 3500–0 Ma with a maximum allowed heating/cooling rate of 10°C/Myr (time step of 10 Myr). The modern surface temperature was allowed to be 0–10°C and the high-temperature starting condition was 400–350°C. White bar in each panel represents the Cryogenian Snowball Earth period from 717–635 Ma. QTQt plotting script is available at: **https://github.com/OpenThermochronology/QTQtPlot**.

**5 Bayesian MCMC inversion tests**

Here we show new models from a thermal history inversion code: `Thermochron.jl` (Keller et al., 2022), that utilizes a (transdimensional) Markov chain Monte Carlo algorithm similar to QTQt. The code is publicly available as a registered Julia package on OSF (https://doi.org/10.17605/osf.io/wq2U5) or Github at https://github.com/OpenThermochronology. Thermochron.jl currently only inverts the zircon (U–Th)/He system but is still in development with other thermochronometers being added. The code utilizes a 1-D Crank-Nicholson finite difference diffusion model and the published Guenthner et al. (2013) ZRDAAM kinetics. We present inversions that implement the same kinetic model variants presented in the Fox et al. manuscript with changes to the amorphous frequency factor. We focus on the canonical ZRDAAM and the low amorphous frequency factor model shown in their figure 5C, since that model exhibited the most apparent differences with respect to their normal ZRDAAM inversion. Inversions are shown with and without a geologic constraint. The usage of geologic/other information were minimized to align somewhat with the models presented in the preprint—yet we wanted to determine if the thermal histories are more consistent overall despite the kinetic model changes. A single Cambrian unconformity was either omitted or enforced in the model at 560 ± 80 Ma between 0–50°C. Cambrian and Ordovician rocks are present in Minnesota and the size of the constraint box is set conservatively small in time when compared to the other models shown without boxes (i.e., those models show cooling to surface temps. over a broader time interval—similar to the model that did not have a Cambrian constraint discussed in McDannell et al. 2022a).

**5.1 Date uncertainty sampling**

We allowed date uncertainties to be resampled (i.e., confidence in the derived uncorrected age but not the total uncertainty on that age), while making the same kinetic model changes as in the Fox et al. paper. They briefly mention age uncertainty treatment in QTQt (lines 268–281) and in that paragraph: "*Either additional uncertainty can be assigned to the measurements by resampling a scaling factor (> 1) that multiplies the input errors. This tends to allow the predicted age-[eU] relationship to pass through the observed data+resampled uncertainty. Or, alternatively, the thermal history can be adjusted to change the predicted age-[eU] relationship to try and ensure that the predictions fit the data, at least to within the error.*" We prefer investigating the data uncertainties because in general little is known a priori about the total size of the errors associated with ZHe dates—except that the observed reproducibility of ZHe dates rarely approaches the analytical precision. Known factors such as U-Th zoning, grain geometry estimation/alpha-ejection correction uncertainty (see Reiners et al., 2017 for summary), and radiation damage zoning cause excess age dispersion (Anderson et al., 2020). Currently, radiation damage models assume uniform kinetics. The model misfit between observed and predicted ages cannot be attributed only to the kinetic model because the kinetic model cannot explain the overdispersion of grains with identical histories and eU. Both kinetic model uncertainty and a wide range of geological and analytical uncertainties contribute to the total misfit between model ages and analytical ages. However, dispersion observed for grains of equivalent eU show that much of the misfit is a result of the latter processes (see Minnesota zircons for an example of this). Resampling the total date uncertainties can accommodate both kinetic and other unknown or poorly characterized sources of dispersion.

We handled date uncertainties differently than in QTQt or in the McDannell et al. (2022a) inversions (the latter utilized the 'scaling factor' mentioned by Fox et al.—except for the MRVT dataset; see text below). The zircon (U–Th)/He date uncertainty (`AnalyticalSigma`) was set to 10% for most grains (50% errors highest eU grains) and the `ModelUncertainty` was set to 25 Myr, which is not well known as it depends on annealing/diffusion parameters and decay constants etc—but it is certainly non-zero. A Simulated Annealing approach (SA; e.g., Kirkpatrick et al. 1983; van Laarhoven and Aarts, 1987) was used to increase the rate at which the Markov chain explores the probability space during burn-in, by adding an additional uncertainty term (`InitialUncertainty`; 35 Myr), which slowly decays to 0 with a decay constant of $\lambda$ over the burn-in period[†]. As a result, SA initially makes it more likely to accept an unfavorable solution, but then slowly decreases the probability of accepting lower likelihood solutions as
* * *
[†]The `AnalyticalSigma` was added in quadrature to the `AnnealingSigma` to yield `Sigma`, which is defined as the total date uncertainty. `Sigma = sqrt(AnalyticalSigma`$^2$` + AnnealingSigma`$^2$`)`; where, `AnnealingSigma = InitialUncertainty * exp(-`$\lambda$`)` `+ ModelUncertainty`. So for example, the oldest input MRVT uncorrected grain age was 770 ± 77 Ma. The starting uncertainty was ± 97.6 Ma [given as: sqrt($(35+25)^2 + 77^2$)] that decayed to ± 80.9 Ma [given as: sqrt($(35*exp(-10)+25)^2 + 77^2$)] by the end of burn-in, which is similar in initial error magnitude to the Empirical Bayes error resampling approach used in McDannell et al. (2022a) for the Minnesota example (i.e., oldest grain age input as 770 ± 100 Ma).

the model space is explored. Since it is necessary to temporarily accept a less favorable solution to escape a local optimum (and ultimately find the global optimum), accepting less likely solutions early in the inversion perhaps counterintuitively accelerates convergence by allowing for a more extensive search of the parameter space for the global optimum. As a result, SA can speed up convergence to the stationary $t$–$T$ path distribution with a shorter burn-in. In contrast, in a standard MCMC inversion without SA, low likelihood solutions have an equal probability of being accepted at any time during the burn-in. Given sufficient burn-in, this also results in a thorough global search but may require a longer burn-in to achieve convergence on the global optimum and thus stationarity.

This SA approach should not be confused either with the Hierarchical Bayes uncertainty resampling currently supported by QTQt, or the Empirical Bayes uncertainty resampling (e.g., Malinverno and Briggs, 2004) that can be applied to ZHe data prior to either QTQt or Thermochron.jl inversions. Hierarchical uncertainty resampling in QTQt allows independent, random scaling of each date error, which will not necessarily assist in convergence on the stationary distribution. Hierarchical and Empirical Bayes resampling may also change the posterior distribution if the date uncertainties are either under- or overestimated. Ideally, Hierarchical and Empirical Bayes resampling both increase the accuracy of the posterior (i.e., make the stationary distribution better reflect reality). Whereas, SA does not increase accuracy but it will help find the posterior distribution more quickly and can be combined with forms of Hierarchical or Empirical Bayes resampling.

[Figure]

**(a)** Low amorphous frequency factor Minnesota inversion without geologic constraints.

[Figure]

**(b)** Low amorphous frequency factor Minnesota inversion with near-surface constraint at 560 ± 80 Ma between 0–50°C (black dashed box).

**Figure 3:** Inversion results for the Minnesota zircons using `Thermochron.jl` and the low amorphous frequency factor kinetics provided in the Fox et al. preprint. The Markov chain was run for 500,000 total iterations with a burn-in of 250,000 iterations. All other models parameters were the same as those in figure 2. We either omitted or enforced a single unconformity constraint in the model at 560 ± 80 Ma between 0–50°C.

**5.2 Inversion results**

Results demonstrate that there are no discernible differences (figs. 2 and 3) for parts of the inverted thermal histories with greater thermochronological sensitivity using the published ZRDAAM kinetics (figs. 2b and 2c) or the low amorphous frequency factor kinetics (figs. 3a and 3b). We recovered essentially the same thermal histories for each kinetic model variant with some expected, but minor, differences in the predicted vs. observed ZHe dates (fig. 4). Thermal histories were accepted that reach surface temperatures in late Precambrian/Cambrian time (figs. 2a and 3a), but their likelihood is lower because the data lack sensitivity to cooler temperatures, and simpler histories explain the data nearly as well. That was also a feature found to some degree in the Minnesota thermal history published in McDannell et al. (2022a; fig. 2C) that disappeared with applied geologic information (see McDannell et al. 2022a; fig. S1). Inadequate burn-in for the model in figure 2a produced a simple linear (but multimodal) high probability $t$–$T$ region that was better resolved with a longer burn-in of 350,000 iterations (fig. 2b). Note that posterior probability behavior similar to this is present in all of the Fox et al. QTQt inversions. The approximate Cambrian surface constraint is necessary to moderate overly simplistic linear cooling posterior $t$–$T$ paths spanning the late Neoproterozic to early Phanerozoic. The Minnesota inversions shown here are able to generally reproduce those shown in McDannell et al. (2022a); here excluding apatite (U–Th)/He data.

The Cambrian geologic constraint improves recovery/resolution of the late Neoproterozoic-early Paleozoic thermal history for both kinetic variants (figs. 2c and 3b). The Phanerozoic portion of the model reproduces the Phanerozoic geologic record for Minnesota (Jirsa et al., 2011). We acknowledge that there are differences in the thermal histories for the pre-1200 Ma history for the canonical ZRDAAM and the low amorphous frequency factor models. The amount of Neoproterozoic heating allowed in the low-frequency-factor models is $\sim$30–40 °C hotter than the conventional ZRDAAM model, but this varies depending on kinetics, burn-in length, and trade-offs in time and temperature; compare figure 2b, figure 2c, and figure 3a near 1000–700 Ma. It may be that the low amorphous frequency factor kinetics produce unrealistically high temperatures required to reset the lowest damage (low-eU) zircons, since the late Paleoproterozoic was the last time MRVT basement underwent local magmatism and metamorphic temperatures of $\sim$300–350°C (e.g., Goldich, 1970; Bauer et al., 2011), although the complex regional geological evolution is still coming into focus (e.g., Southwick, 2014). The zircons are nonetheless thermally reset, regardless of the differences in heating magnitude between models. As Fox et al. stated (lines 301–302): "*The large uncertainties on the parameters controlling helium diffusion in zircon and the dramatic impact this has on temperature sensitivity highlights that this is important to consider.*" The models we have shown here exhibit subtle differences in the recovered thermal histories, but overall they are very similar where the data have $t$–$T$ sensitivity, which is reinforced when a single approximated geologic constraint is added.

**Cryogenian cooling is consistently present in all of the inversions**

While the ZRDAAM kinetic calibration of Guenthner et al. (2013) is not perfect, the conclusions of Fox et al. may be overstated with respect to the effects of kinetic uncertainties on inverted thermal histories. That is not to say that more laboratory diffusion experiments on zircon should not be done—they most certainty should be performed. The authors provide an innovative solution to better incorporate estimates of kinetic model uncertainty into $t$–$T$ inversions and this avenue is definitely worth pursuing in ongoing work. The authors may argue that some of the points addressed herein are outside the scope of their original manuscript, but it is one thing for their paper to discuss a realistic outlook on the precision and accuracy of kinetic models, and another to discuss those concepts for a single deep-time example—in our opinion, this results in a misleading framing of the current issues and their broader applicability. Hopefully this comment will stimulate further conversation on these important topics.

Kind regards,

*K. McDannell*

[Figure]

**(a)** Observed versus model predicted dates with respect to eU for ZRDAAM inversion without geologic constraint. 250,000 iteration burn-in.

[Figure]

**(b)** Observed versus model predicted dates with respect to eU for ZRDAAM inversion without geologic constraint. 350,000 iteration burn-in.

[Figure]

**(c)** Observed versus model predicted dates with respect to eU for ZRDAAM inversion with geologic constraint.

[Figure]

**(d)** Observed versus model predicted dates with respect to eU for low amorphous frequency factor inversion without geologic constraint.

[Figure]

**(e)** Observed versus model predicted dates with respect to eU for low amorphous frequency factor inversion with geologic constraint.

**Figure 4:** Predicted zircon (U–Th)/He date-eU trends for the `Thermochron.jl` inversions. Purple points are input uncorrected data shown with uncertainty (`AnalyticalSigma`*2; for plotting purposes only). Other colored points are the predicted dates from the posterior distribution of the accepted $t$–$T$ paths.

**References**

Anderson, A. J., Hodges, K. V. and van Soest, M. C.: Empirical constraints on the effects of radiation damage on helium diffusion in zircon, Geochim. Cosmochim. Acta, 218, 308–322, doi:10.1016/j.gca.2017.09.006, 2017.

Anderson, A. J., Hanchar, J. M., Hodges, K. V. and van Soest, M. C.: Mapping radiation damage zoning in zircon using Raman spectroscopy: Implications for zircon chronology, Chem. Geol., 538, doi:10.1016/j.chemgeo.2020.119494, 2020.

Bauer, R. L., Bickford, M. E., Satkoski, A. M., Southwick, D. L. and Samson, S. D.: Geology and geochronology of Paleoarchean gneisses in the Minnesota River Valley, in GSA Field Guides: Archean to Anthropocene: Field Guides to the Geology of the Mid-Continent of North America, vol. 24, edited by J. D. Miller, G. J. Hudak, C. Wittkop, and P. I. McLaughlin, pp. 47–62., 2011.

Flowers, R. M., Ketcham, R. A., Shuster, D. L. and Farley, K. A.: Apatite (U–Th)/He thermochronometry using a radiation damage accumulation and annealing model, Geochim. Cosmochim. Acta, 73(8), 2347–2365, doi:10.1016/j.gca.2009.01.015, 2009.

Flowers, R. M., Ketcham, R. A., Macdonald, F. A., Siddoway, C. S. and Havranek, R. E.: Existing thermochronologic data do not constrain Snowball glacial erosion below the Great Unconformities, Proc. Natl. Acad. Sci., 119(38), e2208451119, doi:10.1073/pnas.2208451119, 2022.

Fox, M. and Shuster, D. L.: The influence of burial heating on the (U-Th)/He system in apatite: Grand Canyon case study, Earth Planet. Sci. Lett., 397, 174–183, doi:10.1016/j.epsl.2014.04.041, 2014.

Gallagher, K.: Transdimensional inverse thermal history modeling for quantitative thermochronology, Journal of Geophysical Research: Solid Earth, 117(B2), doi:10.1029/2011JB008825, 2012.

Gallagher, K.: Comment on "A reporting protocol for thermochronologic modeling illustrated with data from the Grand Canyon" by Flowers, Farley and Ketcham, Earth and Planetary Science Letters, 441, 211–212, doi:10.1016/j.epsl.2016.02.021, 2016.

Gallagher, K.: Comment on "Discussion: Extracting thermal history from low temperature thermochronology/A comment on the recent exchanges between Vermeesch and Tian and Gallagher and Ketcham", by Paul Green and Ian Duddy, Earth Science Reviews, https://doi.org/10.1016/j, Earth-Science Reviews, 216, 103549, doi:10.1016/j.earscirev.2021.103549, 2021.

Gautheron, C., Tassan-Got, L., Barbarand, J. and Pagel, M.: Effect of alpha-damage annealing on apatite (U-Th)/He thermochronology, Chem. Geol., 266(3–4), 157–170, doi:10.1016/j.chemgeo.2009.06.001, 2009.

Ginster, U., Reiners, P. W., Nasdala, L. and Chanmuang, C.: Annealing kinetics of radiation damage in zircon, Geochim. Cosmochim. Acta, 249, 225–246, doi:10.1016/j.gca.2019.01.033, 2019.

Goldich, S. S., Hedge, C. E. and Stern, T. W.: Age of the Morton and Montevideo gneisses and related rocks, southwestern Minnesota, Bull. Geol. Soc. Am., 81(12), 3671–3696, doi:10.1130/0016-7606(1970)81[3671:AOTMAM]2.0.CO;2, 1970.

Guenthner, W. R., Reiners, P. W., Ketcham, R. A., Nasdala, L. and Giester, G.: Helium diffusion in natural zircon: radiation damage, anisotropy, and the interpretation of zircon (U-Th)/He thermochronology, Am. J. Sci., 313(3), 145–198, doi:10.2475/03.2013.01, 2013.

Guenthner, W. R.: Implementation of an Alpha Damage Annealing Model for Zircon (U-Th)/He Thermochronology With Comparison to a Zircon Fission Track Annealing Model, Geochemistry, Geophys. Geosystems, 22(2), doi:10.1029/2019GC008757, 2021.

Guo, H., Zeitler, P. K., Idleman, B. D., Fayon, A. K., Fitzgerald, P. G. and McDannell, K. T.: Helium diffusion systematics inferred from continuous ramped heating analysis of Transantarctic Mountains apatites showing age overdispersion, Geochimica et Cosmochimica Acta, 310, 113–130, doi:10.1016/j.gca.2021.07.015, 2021.

Jirsa, M. A., Boerboom, T. J., Chandler, V. W., Mossler, J. H., Runkel, A. C. and Setterholm, D. R.: S-21 Geologic Map of Minnesota-Bedrock Geology, Minnesota Geological Survey, https://hdl.handle.net/11299/101466, 2011.

Johnson, J. E., Flowers, R. M., Baird, G. B. and Mahan, K. H.: "Inverted" zircon and apatite (U–Th)/He dates from the Front Range, Colorado: High-damage zircon as a low-temperature (<50 °C) thermochronometer, Earth Planet. Sci. Lett., 466, 80–90, doi:10.1016/j.epsl.2017.03.002, 2017.

Keller, C. B., McDannell, K. T., Guenthner, W. and Shuster, D. L.: Thermochron.jl: Open-source time-Temperature inversion of thermochronometric data, doi: 10.17605/osf.io/wq2U5, 2022.

Ketcham, R. A., Guenthner, W. R. and Reiners, P. W.: Geometric analysis of radiation damage connectivity in zircon, and its implications for helium diffusion, Am. Mineral., 98(2–3), 350–360, doi:10.2138/am.2013.4249, 2013.

Ketcham, R. A.: Fission-Track Annealing: From Geologic Observations to Thermal History Modeling, in Fission-Track Thermochronology and its Application to Geology, edited by M. G. Malusá and P. G. Fitzgerald, pp. 49–75, Springer, Cham., 2019.

Kirkpatrick, S., Gelatt, C. D. and Vecchi, M. P.: Optimization by simulated annealing, Science, 220(4598), 671–680, doi:10.1126/science.220.4598.671, 1983.

Mackintosh, V., Kohn, B., Gleadow, A. and Tian, Y.: Phanerozoic Morphotectonic Evolution of the Zimbabwe Craton: Unexpected Outcomes From a Multiple Low-Temperature Thermochronology Study, Tectonics, 36(10), 2044–2067, doi:10.1002/2017TC004703, 2017.

Malinverno, A. and Briggs, V. A.: Expanded uncertainty quantification in inverse problems: Hierarchical Bayes and empirical Bayes, Geophysics, 69(4), 1005-1016, doi:10.1190/1.1778243, 2004.

McDannell, K. T., Schneider, D. A., Zeitler, P. K., O'Sullivan, P. B. and Issler, D. R.: Reconstructing deep-time histories from integrated thermochronology: An example from southern Baffin Island, Canada, Terra Nov., 31(3), 189–204, doi:10.1111/ter.12386, 2019.

McDannell, K. T. and Flowers, R. M.: Vestiges of the Ancient: Deep-Time Noble Gas Thermochronology, Elements, 16(5), 325–330, doi:10.2138/gselements.16.5.325, 2020.

McDannell, K. T. and Keller, C. B.: Cryogenian glacial erosion of the central Canadian Shield: The "late" Great Unconformity on thin ice, Geology, 50, doi:10.1130/G50315.1, 2022.

McDannell, K. T., Keller, C. B., Guenthner, W. R., Zeitler, P. K. and Shuster, D. L.: Thermochronologic constraints on the origin of the Great Unconformity, Proc. Natl. Acad. Sci., 119(5), e2118682119, doi:10.1073/pnas.2118682119, 2022a.

McDannell, K. T., Keller, C. B., Guenthner, W. R., Zeitler, P. K. and Shuster, D. L.: Reply to Flowers et al.: Existing thermochronologic data constrain Snowball glacial erosion below the Great Unconformity, Proc. Natl. Acad. Sci., 119(38), e2209946119, doi:10.1073/pnas.2209946119, 2022b.

Powell, J., Schneider, D., Stockli, D. and Fallas, K.: Zircon (U-Th)/He thermochronology of Neoproterozoic strata from the Mackenzie Mountains, Canada: Implications for the Phanerozoic exhumation and deformation history of the northern Canadian Cordillera, Tectonics, 35(3), 663–689, doi:10.1002/2015TC003989, 2016.

Recanati, A., Gautheron, C., Barbarand, J., Missenard, Y., Pinna-Jamme, R., Tassan-Got, L., Carter, A., Douville, E., Bordier, L., Pagel, M. and Gallagher, K.: Helium trapping in apatite damage: Insights from (U-Th-Sm)/He dating of different granitoid lithologies, Chem. Geol., 470, 116–131, doi:10.1016/j.chemgeo.2017.09.002, 2017.

Reiners, P. W., Carlson, R. W., Renne, P. R., Cooper, K. M., Granger, D. E., McLean, N. M. and Schoene, B.: The (U–Th)/He system, in Geochronology and Thermochronology, pp. 291–363, John Wiley & Sons., 2017.

Southwick, D. L.: Reexamination of the Minnesota River Valley Subprovince with emphasis on Neoarchean and Paleoproterozoic events, Minnesota Geological Survey, Report of Investigations 69, 52 p., 2014.

van Laarhoven, P. J. M. and Aarts, E. H. L.: Simulated annealing, in Simulated Annealing: Theory and Applications, edited by P. J. M. van Laarhoven and E. H. L. Aarts, pp. 7–15, Springer Netherlands, Dordrecht., 1987.

Willett, S. D.: Inverse modeling of annealing of fission tracks in apatite 1: A controlled random search method, Am. J. Sci., 297(10), 939–969, doi:10.2475/ajs.297.10.939, 1997.

Willett, C. D., Fox, M. and Shuster, D. L.: A helium-based model for the effects of radiation damage annealing on helium diffusion kinetics in apatite, Earth Planet. Sci. Lett., 477, 195–204, doi:10.1016/j.epsl.2017.07.047, 2017.

---

## Author Comment (AC2)

Reviewer: Kalin McDannell

We appreciate the detailed comment provided by Kalin McDannell on our paper. We have reframed some of the introduction and key points in our manuscript to highlight that this is indeed a general problem and certainly not unique to deep time thermochronomety. We would like to take this opportunity to clarify some points, but many of these points should really be discussed in a separate paper that goes into more detail. We have broken this response down into the same sections as the original review.

1. Kinetic model uncertainty

We appreciate that the uncertainties that we have highlighted are not unique to the Great Unconformity debate. We have added text to highlight that this is important for other applications. We have also restructured the introduction to begin with a section on general thermochronometric methods. However, much of the work on deep time thermochronology relies on interpreting data that have been influenced by radiation damage. This is why we focused on this interesting point.

2. Parameter correlations

We think we have done a better job at accounting for these parameter correlations in our revised manuscript. Now we have a modified version of QTQt that explicitly accounts for uncertainties in the damage model. We have also approximated the distribution as 4D guassian that allows us to explicitly define a covariance and correlation matrix. These clearly highlight the model correlations.

3 .The effects of additional constraints on inversions results?

We have removed these inversions because they have caused so much controversy and detract from our very simple message. Instead we show a new QTQt model where we have reduced the uncertainties on the ages. We were unable to replicate the thermal histories inferred from in the comment using the reported data uncertainties. We modified the length of the burn in but this had no effect on the inferred thermal history. This work was carried out by Kerry Gallagher, who is an expert in thermal history modelling.

4. Posterior probabilities

See previous point.

4.1 MCMC burn-in and ZHe uncertainties

K. Gallagher ran the QTQt models with the quoted errors from the original publications. Changing the burn in length made no appreciable difference. Instead, we think the issue may be related to the data uncertainty. We have attempted to address this here and we are now able to recreate a GU-type thermal history.

4.2 Timing of cooling

It is not clear what the reviewer is referring to here. We had used the point in time at which the green colour first appears as this represents when the path is well resolved. This is when the thermal history is resolved, we are saying nothing about when a period of cooling is resolved.

4.3 Continuous spread of ages as a function of eU

Yes, we would argue against binning data in general. The binning of the data potentially removes valuable information that might guide future research. However, binning may be required in certain cases.

5. Bayesian MCMC inversion tests

These models are very interesting and highlight the importance of testing model parameter values. We thank Kalin for highlighting this and we have gone to great lengths to fully explore this in the new QTQt section of our paper.

---

## Author Comment (AC3)

Reviewer: Kip Hodges

*Our responses are to the reviewer's comments are shown in italics.*

Review of Fox et al., "Origin of Great Unconformity Obscured by Thermochronometric Uncertainty"

*We thank Kip for his helpful and insightful review.*

In this manuscript, Fox and co-authors present a detailed and useful discussion regarding the effects of uncertainties related to the Zircon Radiation Damage and Annealing Model (ZRDAAM) on thermal history models based on zircon (U-Th)/He data. They go on to suggest that these effects complicate thermochronologic models that have been used to constrain plausible causes for the Great Unconformity at the end of the Precambrian.

After a brief review of the reasons why some sort of ZRDAAM-like model is necessary to reasonably interpret zircon (U-Th)/He data, the authors correctly note that the original ZRDAAM formulation of Guenthner and co-workers (Guenthner et al., 2013), based on helium diffusion experiments on variably damaged zircons, has shortcomings that make it difficult to realistically propagate uncertainties in two experimentally derived helium diffusion parameters – the pre-exponential constant $D_o$ and activation energy $E_a$ – into ZRDAAM models and onward into thermal history models. In order to do this more quantitatively, they first present a revised ZRDAAM formulation in Section 3 of the manuscript. They show that diffusion parameters derived from experimental data using the new formulation mostly agrees with the diffusion parameters adopted by Guenthner et al. for a highly damaged ("amorphous") crystal, but differ markedly for an "undamaged" crystal (Figures 1 and 2). They attribute this to difficulties in estimating $D_o$ and $E_a$ independently using the approach of Guenthner et al. Most importantly, however the authors show that the uncertainties on derived parameters are quite large, much larger than the thermochronology community generally assumes.

*Yes, these uncertainties are quite large. However, the parameters are also strongly correlated. This means that the impact on the estimates is not quite a large as might be expected, if the parameter correlation structure is accounted for.*

Using the revised ZRDAAM formulation, Fox et al. then proceed to explore how such large uncertainties in radiation damage and annealing model parameters might affect thermal history models calculated using the QTQt software package of Gallagher et al. (Gallagher, 2012). Unfortunately, direct propagation of such uncertainties into inverse modeling of thermal histories using QTQt is computationally impractical (lines 302-305), but the authors present the results of efforts using simplified models showing that uncertainties of up to several hundreds of million years in the estimated timing of a cooling event should not be unexpected (!). They go on to point out, specifically, that these results have dramatic implications for studies such as those being conducted by several groups on the Great Unconformity.

*Yes, we argued that this has major implications for when the thermal history is resolved. The impact on when specific events are resolved along a thermal history is a bit more complicated. This is highlighted in the comment by Kalin McDannell. We have dealt with this point more clearly in our revised manuscript.*

I enjoyed reading this well-written contribution and feel it is an important step toward developing a robust appreciation of how confident we should be in the results of thermal history modeling studies based on datasets for which either zircon or apatite (U-Th)/He radiation damage modeling is required. I found no fault in the mathematics presented here and I trust the authors to have done the modeling carefully. I hope, however, that these findings are not used by non-specialists to conclude that thermal history modeling using thermochronologic data is practically useless because the uncertainties are so large. Fox et al. have tried a bit to guard against that, but they might be more explicit about that in the Implications section. The principal lessons I think we should learn are: 1) we should not overinterpret every wiggle in a modeled time-temperature path as having geologic significance; 2) we should not overestimate the precision with which we know the timing of specific cooling (or reheating) events given the imprecision of derived diffusion and annealing parameters; and 3) we should all continue to work hard to improve our understanding of the impact of radiation damage on diffusion parameters through a combination of experimental and empirical studies. I was happy to see the nod toward also using natural laboratories to address these issues moving forward (lines 319-322). In fact, led by Alyssa Anderson, our research group at Arizona State performed such a study (Anderson et al., 2017), concluding that the Guenthner et al. formulation failed to predict ZHe closure behavior consistent with QTQt models of an independent multichronometer dataset for the McClure Mountain syenite of Colorado. However, Anderson and co-workers preferred to interepret this inconsistency as a consequence of variable and complex radiation damage zoning in individual McClure Mountain zircons rather than uncertainties in radiation damage. Certainly, however, uncertainties in the original ZRDAAM model might have played a role, as the Fox et al. work indicates. (It should be noted that, while the accuracy of the cooling history we derived has been disputed based on geological interpretations (Weisberg et al., 2018), the alternative proposed cooling history still remains inconsistent with the ZrnHe closure behavior predicted by the original ZRDAAM model (Anderson et al., 2018).) Our study and the current manuscript together remind us that the effects of radiation damage on ZHe closure behavior of ZHe is extremely complex and dependent on both the parameters we choose for modeling and the idiosyncracies of specific zircons. In such a climate of uncertainty, interpretive caution is advised.

*We have included these citation in our revised manuscript and thank Kip for highlighting these to us. It is also nice to see the three lessons we should learn so clearly articulated. This is exactly what we hope people take from the analysis.*

*– Kip Hodges*
*References*
*Anderson, A. J., Hodges, K. V., and van Soest, M. C., 2017, Empirical constraints on the effects of radiation damage on helium diffusion in zircon: Geochimica et Cosmochimica Acta, v. 218, p. 308-322.*

*Anderson, A. J., Hodges, K. V., and van Soest, M. C., 2018, Comment on 'Distinguishing slow cooling versus multiphase cooling and heating in zircon and apatite (U-Th)/He datasets: The case of the McClure Mountain syenite standard' by Weisberg, Metcalf, and Flowers: Chemical Geology, v. 498, p. 150-152.*

*Gallagher, K., 2012, Transdimensional inverse thermal history modeling for quantitative thermochronology: Journal of Geophysical Research, v. 117, p. 2156-2202.*

*Guenthner, W. R., Reiners, P. W., Ketcham, R. A., Nasdala, L., and Giester, G., 2013, Helium diffusion in natural zircon: Radiation damage, anisotropy, and the interpretation of zircon (U-Th)/He thermochronology: American Journal of Science, v. 313, no. 3, p. 145- 198.*

*Weisberg, W. R., Metcalf, J. R., and Flowers, R. M., 2018, Distinguishing slow cooling versus multiphase cooling and heating in zircon and apatite (U-Th)/He datasets: the case of the McClure Mountain syenite standard: Chemical Geology, v. 485, p. 90-99.*

---

## Author Comment (AC4)

Reviewer: Rebecca Flowers

*Our responses are to the reviewer's comments are shown in italics.*

Review of Fox et al., Origin of Great Unconformity Obscured by Thermochronometric Uncertainty, Geochronology

*We thank Prof. Flowers for this detailed and helpful review. We have modified the text where suggested and this has improved the framing of the problem and also the ongoing debate. We have attempted to satisfy all reviewers with our modifications. These are listed in the general response we have provided. Due to the number of changes requested by the reviewers, we have not tracked changes.*

*The core of the discussion focuses on whether the thermochronometry alone can be used to constrain cooling histories through time or whether constraints are required. It is our belief that it is very important to establish what the data can and can not resolve. It is useful to show what a solution might look like without any constraints as in many cases, the constraints are controversial or complicated. Through linking samples in space using concepts like the AER, or grouping samples together, more thermochronometric data is used to infer a solution. This should perhaps be attempted before incorporating additional constraints. It is also important to highlight that incorporating data from different thermochronometric systems will also provide greater resolution.*

*Our goal was to simply explore the importance of uncertainty in damage parameters. To isolate this, we attempted to infer a thermal history, only using the data. Our model was too simple, however, the best fitting model does approach an acceptable GU thermal history. To ensure that we are able to produce this thermal history, we have decreased the uncertainty on the data. In this way, we have produced a thermal history that is consistent with the GU but have also been able to explore the importance of the uncertainty.*

This contribution is focused on better constraining the uncertainty associated with the zircon radiation damage accumulation and annealing kinetic model (ZRDAAM), and then evaluating how this uncertainty influences the ability to resolve thermal histories associated with development of the Great Unconformity. The manuscript begins by reviewing the existing ZRDAAM calibration, presents a new calibration based on the same kinetic dataset as used in the original ZRDAAM, uses a new approach to constrain ZRDAAM uncertainties, carries out inversion modeling of a dataset to explore the influence of ZRDAAM kinetic uncertainties on data interpretations, and concludes with some implications regarding the resolution of (U-Th)/He datasets in deep time.

This is a useful and informative contribution. I agree that there's opportunity to use the current Great Unconformity debate as motivation to improve kinetic model calibrations, which in turn can improve our ability to address problems like the origin of this feature. This paper seems to approach the Great Unconformity problem from the direction of evaluating whether the thermochronologic data alone can resolve thermal histories in deep time. I completely agree that this absolutely is not possible without integration with other types of information, like that from geologic data.

I provide a variety of suggestions below that I think could be used to further strengthen the paper. Most importantly I highlight (described further below): 1) the need to more clearly and completely explain the new calibration, 2) that it's essential to include the Great Unconformity in any model that is supposed to explain the Great Unconformity, 3) that both the data being modeled and the predictions from the inversion models should be plotted, and 4) one aspect of the implications section that could be more valuable if expanded.
Abstract

Lines 13-14: "...about the origin of the Great Unconformity, a global erosional event that represents a period of almost a billion years at the end of the Precambrian."
Suggest modifying this sentence. The Great Unconformity is a geologic feature, not an event. This sentence also asserts that it represents a global erosional event that represents a billion years – but the debate is actually about the timing and duration and whether it represents a global erosional event or not.
Introduction

*Corrected.*

Line 33: Keller et al. (2019) isn't appropriate to cite in this sentence because there are no thermochronologic data presented or interpreted in that contribution. Also suggest adding an e.g., before the refs, or adding more refs, because there are more papers that have used thermochronologic datasets to make interpretations about the Great Unconformity than those cited here.

*We have modified the introduction and this has been accounted for.*

Lines 35-36. "...in the Grand Canyon, the erosional event spans from circa 1200 to 250 million years...". This is the amount of missing time. The erosional event didn't span this entire interval. Also should use an e.g., for the reference, because many have noted this missing magnitude of time in the Grand Canyon.

*Corrected.*

Lines 38-40. "This approach highlighted that erosion rates increased across the North American craton during Neoproterozoic glaciation, supporting the hypothesis that...". It would be more appropriate to replace the word "highlighted" with "inferred". None of the datasets used in that paper require that erosion increased during the Snowball glaciation, as shown in the comment by Flowers et al. (2022). This is true even when one does not take into account kinetic model uncertainties, which I agree further increase the uncertainty in the inferred tT paths, and are the focus of this contribution.

*Corrected.*

Line 46: "...concepts of geochronology". Also involves concepts of diffusion. Suggest modifying to "...geochronology and diffusion."

*Corrected.*

Line 49: missing a word here, should be "diffusive loss".

*Corrected.*

Lines 43-51: Suggest emphasizing in this paragraph that the dates represent a time-integrated thermal history. This paragraph focuses only on simple exhumation scenarios, which differ from the deep-time studies discussed in this paper, that include multiple burial and erosion events.

*Corrected.*

Line 57: suggest rewording to "the transition from open-system to closed-system" given the phrasing of the previous sentences.

*We have kept this sentence because it is quite general.*

Line 59: Appropriate to cite Ketcham et al. (2013) here, along with Guenthner et al., 2013.

*Cited.*

Line 67. "Both those in favour of a glacial (McDannell et al., 2022) and tectonic (Flowers et al., 2020) origin..."

To be more accurate, suggest modifying to "Both those in favor of a globally synchronous glacial origin (McDannell et al., 2022) and regionally diachronous tectonic origins (Flowers et al., 2020)..." The text has framed the problem here and elsewhere as a debate over a single origin for this feature – but a critical aspect of the debate is whether or not there is indeed a singular origin or if there are multiple origins.

*Corrected.*

Lines 69-77. This is a great description, and also nicely highlights that annealing at higher temperatures after damage accumulation at low temperatures affects the retentivity and the date. It's this element that could be useful to bring into the geological framing in lines 43-51.

*Thank you! We want to keep the general structure and keep the intro to thermochron as concise as possible. With all the shifting of the intro, we have hopefully achieved this and feel like it is acceptable to focus on the problem here.*

Line 79. Suggest changing RDAAM to ZRDAAM, as ZRDAAM was defined in previous paragraph. RDAAM refers to a widely used apatite radiation damage accumulation and annealing model, which isn't discussed in this paper.

*Corrected.*

Lines 81-85. "Parts of this history were reported to within less than 10 degrees between 700 and 250 Ma and then again from 15-7 Ma. It is unclear whether the data really provide such tight constraints on temperatures in the past..."
- It's clear that the data don't provide such tight constraints on the temperatures in the past, because the model of their Figure 4 says that when the Great Unconformity developed that the basement was at 200 +/- 10 °C (so, 7-10 km deep in the crust, depending on assumptions), when we know unequivocally that the rocks were at the surface during deposition of the Cambrian sandstone on the basement to define the Great Unconformity. This model isn't set up to require that the Great Unconformity be part of the model. See Peak et al., 2022. This doesn't seem like a good example to use here.

*We have kept this example here to highlight that sometimes thermal histories appear to be too well resovled. Because our focus is on the importance of damage model uncertainties, this is a really nice example to highlight as our damage model would increase the uncertainties.*

Lines 87-88. "...known amounts of radiation damage." Suggest being more cautious here. Instead of saying "known amounts of damage", could say "...well-constrained amounts of radiation damage." For the Sri Lankan zircon, which I agree was a good sample to use in the Guenthner et al. study, the assumption of rapid cooling at 440-420 Ma to estimate radiation damage seems reasonable, but this is a long timescale with some uncertainty in the history. And even if one acquires Raman data, there is currently ambiguity in how to translate that data into the type(s) of damage in the crystal that matter for He diffusion.

Lines 94-95. "We show that natural variability in radiation damage annealing parameters causes ZHe ages to be dispersed even for crystals..." Delete "annealing"? As I understand it, this paper doesn't address variability and uncertainty in radiation damage annealing kinetics?

*Corrected.*

The existing calibration of the radiation damage and annealing model
Line 123. "...at a specific cooling rate." And assumed grain size.

*Corrected.*

Line 139. "..between the model parameters." Helpful to specify here that you mean zDo and zEa.

*Corrected.*

Lines 143-144. "However the accuracy of this model has only been assessed by looking at general trends in model predictions." Do you mean in Guenthner et al.? Or in the variety of studies that have aimed to explain data using this kinetic model? Provide some references here?

*We have only cited the Guenthner paper here as these data are the best to constrain the parameters and this is the only time the model has been compared to the data in a quantitative way.*

Lines 125-146. Clear explanation of the Guenthner model.

*Thank you!*

It would be worth explicitly noting somewhere that this contribution does not address uncertainties in the annealing model.

*We have highlighted this point at the end of the paragraph. Lines 144.*

A new calibration of the zircon radiation damage and annealing model
This section is the crux of the paper and I think that more explanation is needed. I believe that I understand the Guenthner calibration, but I don't understand how the end member values for this new calibration (and thus their associated uncertainties) were obtained here based on the explanation provided. If what is presented here is indeed an improved approach as the authors are clearly arguing, then this is exciting and it should be explained with enough clarity so that those who generate the datasets used for model calibrations can understand and apply this method. If the authors would prefer not to lengthen the main text, then the appendix would be a great place for an extended description.
Lines 194-196. From the Figure 1A and 1B plots, it's not obvious that the revised calibration predicts the observed data better than the original calibration. It would be helpful to add some text here that explains what features of this plot that the reader should focus on to see this.

*We have refined this plot by adding colour that reflects total alpha dose. We have also stressed that both models do not fit all the variability in the data. Our aim to assess the uncertainties, not provide a new damage model.*

Figure 1. If I understand correctly, the purpose of this figure is to show how well the Arrhenius data for each sample used in the ZRDDAM calibration are fit with the original calibration and the ZRDAAM calibration of this paper. Right now, it's difficult to make this comparison across the two plots. This probably could be done more successfully by including a separate Arrhenius plot for each sample, labelling each Arrhenius plot with the sample name, and on the same plot including the diffusion kinetic data, the prediction from the original ZRDAAM calibration, and the prediction from the ZRDAAM calibration for that sample.

*This is not the goal of our analysis. It would also make too many figures and complicate the simple message we wish to highlight. This would be the core of a new paper that attempts to improve the fit the to the data with a new parameterization.*

Lines 196-203 and Figure 2. Could you please explain more fully and clearly how these histograms were calculated. Again, this is the crux of the paper, since this calibration and

the associated uncertainties are then applied to draw the main conclusions of this contribution.

*We have improved the overall description of the model and the parameters.*
Propagating model uncertainties
Lines 222-223. "Note, this implementation of the model has been used previously (Tripathy-Lang et al., 2015). It sounds like this model calibration was presented seven years ago in previous work? To what extent was it described there? This should be explained in the introduction.

*We have clarified that this refers to the numerical methods.*

Lines 224. "...an effective U concentration ([eU = [U] +0.24[Th]; Gastil et al., 1967))." I hadn't encountered this paper before, which interestingly uses eU to refer to "equivalent uranium" (although not effective uranium). That paper doesn't include an eU equation. Cooperdock et al. (2019) should be cited here for the equation, (eU = U + 0.238Th). Flowers et al., 2022 also lays out the equations for eU.

*Added references here.*

Line 241. "...and the data from McDannell et al. (2022)". Miltich (2005) should be cited for the data. McDannell et al. mined the Minnesota data from the Miltich (2005) undergraduate honors thesis without publishing them.

*Added the references here.*

Line 242. "...for the sample "Minnesota"... These are actually multiple samples, not a single sample.

*Corrected and clarified.*

Lines 241-242 and Figure 5. "For this reason, we use only the ZHe data and do not incorporate additional constraints."
- This simulation is supposed to explain the Great Unconformity, but the highest probability time-temperature paths aren't at surface conditions when the Great Unconformity developed. In the final comparative statement that compares the three panels, the caption states "The overall patterns are very similar, but the apparent resolution is different, resulting in different geological conclusions." To me, the geological conclusion here should be that all of these model results are geologically meaningless because the highest probability tT paths violate the Great Unconformity, so all three models should be discarded. Could the authors either include the Great Unconformity in the models or better articulate why the Great Unconformity isn't honored in a model that is supposed to explain the Great Unconformity? This is now a repeated characteristic of many published QTQt models that are supposed to reproduce the Great Unconformity.

*We have redone the analysis and the GU is reproduced to some degree now. The discussion about whether to include constraints or not is at the core of thermal history modelling and requires a separate discussion, not a paragraph in a paper on damage models.*

Figure 5. Please plot the data being modeled here and show how well the tT paths fit the observed data – for example, by making date-eU plots of observed vs. modeled data. See Gallagher (2016).

*The data are well predicted by the model. These plots can be seen in the original publications.*

Implications
Lines 264-266. "In turn, it may be challenging to resolve cooling histories sufficiently to attribute the Grat Unconformity to Cryogenic Glaciations (McDannell et al. 2022) or geodynamic process related to the break-up of Gondwana (Flowers et al., 2020)." Yes, agreed that it's not currently possible to resolve cooling histories at this level with the "thermochronologic data alone" even when not considering uncertainties in kinetic models. This is why it is essential to integrate other types of information into models, such as geologic data. Flowers et al. (2020) didn't argue that we could resolve the cooling histories sufficiently without other information, as implied in this sentence. Perhaps you could modify to remove that implication?

*Corrected.*

Lines 268-281, and final sentence in this paragraph: "For example, McDannell et al. (2022)'s results for Pikes Peak highlight how models that ignore overdispersion appear to resolve a 700 Ma cooling signature, which is smoothed out when the overdispersion is effectively reduced by adding excess uncertainty on some of the data."
- There are interesting and important points made in this paragraph about how uncertainties can be accounted for in QTQt and the associated influence on the inferred tT paths. The final sentence that makes a vague reference to a figure in McDannell et al. (2022). It would be great if the authors would add a figure in this paper that helps to illustrate the valuable points being made here rather than vaguely referring to a figure in that paper. Alternatively, this paragraph could be eliminated.

*We have kept this to highlight the discussion. We tried drafting a new figure but were unable to make an effective figure.*

- The Pikes Peak models in McDannell et al. (2022) also violate the Great Unconformity relationship, as noted elsewhere about other published models.
- If this model is discussed, the I feel that it also is important to cite our 2022 comment on this paper. In that comment we show that entirely different tT paths, not captured in the McDannell et al. models, can explain the data.
Lines 283-314. This is a nice illustration of the dispersion expected in real datasets, and provides some of the rationale for the binning into synthetic grains as done by many who simulate (U-Th)/He data.

Code Availability. It would be appropriate to put the codes used for the calculations in this paper in a supplement and available for download so that others can reproduce these results and apply the approach to other kinetic datasets. This is now done so easily that it no longer seems appropriate to require an email to the authors to obtain the code.

*QTQt is not available online. We are also attempting to do this same analysis with different diffusion datasets.*

I enjoyed reading this contribution and hope that the authors find these comments helpful to further strengthen the manuscript.

Becky Flowers, CU Boulder

References

Cooperdock, E.H.G., Ketcham, R.A., and Stockli, D.F., 2019, Resolving the effects of 2-D versus 3-D grain measurements on (U-Th)/ He age data and reproducibility: Geochronology, v. 1, p. 17–41, https://doi .org /10 .5194 /gchron-1-17-2019.

Flowers, R.M., Ketcham, R.A., Macdonald, F.A., Siddoway, C.S., Havranek, R.E., 2022, Existing thermochronologic data do not constrain Snowball glacial erosion below the Great Unconformities: Proceedings of the National Academy of Sciences, Letter to the Editor, v. 119, No. 38, https://doi.org/10.1073/pnas.2208451119.

Flowers, R.M., Zeitler, P.K., Danišík, M., Reiners, P.W., Gautheron, C., Ketcham, R.A., Metcalf, J.R., Stockli, D.F., Enkelmann, E., and Brown, R.W., 2022, (U-Th)/He chronology: Part 1. Data, uncertainty, and reporting: Geological Society of America Bulletin special volume on the Reporting and Interpretation of Geochronologic data, https://doi.org/10.1130/B36266.1.

Ketcham, R.A., Guenthner, W.R., and Reiners, P.W., 2013, Geometric analysis of radiation damage connectivity in zircon, and its implications for helium diffusion: The American Mineralogist, v. 98, p. 350–360, https://doi .org /10 .2138 /am .2013 .4249.

L. Miltich, 2005, Low temperature cooling history of Archean gneisses and Paleoproterozoic granites of southwestern Minnesota. Undergraduate thesis, Carleton College, Northfield, MN.

Peak, B.A., Flowers, R.M., Macdonald, F.A., and Cottle, J.M., 2022, Forum, Reply to Comment on: Zircon (U- Th)/He thermochronology reveals pre-Great Unconformity paleotopography in the Grand Canyon region: 50 (3): e544, https://doi.org/10.1130/G49965Y.1.

---

## Author Comment (AC6)

Reviewer: Willy Guenthner

*Our responses are to the reviewer's comments are shown in italics.*

The manuscript by Fox and co-authors re-examines certain aspects of a commonly used damage-diffusivity model for the zircon (U-Th)/He system (ZRDAAM). The authors highlight that the model, as originally derived and defined by Guenthner et al. (2013), currently lacks information on kinetic uncertainty, which could in turn influence the thermal history modeling-based outcomes that rely upon ZRDAAM. In particular, the authors argue that the lack of kinetic uncertainty could create over-interpretation of and/or overconfidence in thermal history model output as applied to a recent discussion surrounding the origin of the Great Unconformity erosion surface. I found this contribution to be of importance for three reasons: 1) it stresses the need to incorporate uncertainties into the kinetic models used by many authors and thermal history modeling software packages, 2) it shows how large zircon He datasets (~40 grains) are needed to refine and extract meaningful time-temperature information, and 3) it recognizes that ZRDAAM is a work in progress that could be strengthened by further He diffusion kinetic work. In sum, I think this manuscript is a solid contribution to ongoing discussions surrounding thermal history modeling and thermochronometric data interpretation. However, given the authors stated goals of exploring calibration issues with ZRDAAM, I think the article does not go far enough in investigating multiple aspects and nuances of the current ZRDAAM calibration. My recommendation is major revisions, which should focus on expanding the scope of the current work. I have several general comments that I would like to see the authors address before this article is published.

*We thank Willy Guenthner for his detailed review and we have addressed as many points as possible. It is important to note that we are not really interested in producing a new damage model – we are simply interested in propagating uncertainties. Points about changing the end member crystal are useful to consider but this requires a completely different model. Furthermore, the approach we develop here can easily be modified to constrain parameters in any damage model. Our goal is to use the diffusion data directly and not rely on an intermediate regression as this ignores correlations in the data.*

*It should also be noted that we don't think that the endmember crystal should really ever be a real crystal. What we have done is attempted to fit all the diffusion data with the same model. In this way, all the data contribute to determining the diffusion kinetics of the end member crystals. We think this is preferable to using something like Mud Tank because then the other crystals and diffusion experiments would not be utilised.*

First, the authors focus heavily on the theoretically pristine endmember zircon, and the fact that this value is an extrapolation that, correctly, could have a large degree of uncertainty to it. It is difficult in the current preprint to determine how important this endmember is in the context of real datasets. Let me expand. When the ZRDAAM was derived, a proposed "zero-damage" grain was used to make the model setup more straight-forward, and to account for potential scenarios where users may want to model thermal histories at very short time scales. But nearly all grains, especially those used for deep-time problems, will very quickly (within ~5 myr for a grain with 500 ppm eU) accumulate damage levels in excess of the Mud

Tank sample, for which empirical results do exist. So perhaps the challenge with the current ZRDAAM is less, what should we do for a zero-damage endmember, but rather, should we instead be using a sample with known kinetics as our endmember? Would the Mud Tank kinetics therefore be more appropriate here as a realistic endmember, and if so, does that reduce some of the variation in thermal history model output? Secondarily, there are both molecular dynamics (Reich et al., 2007; Saadoune et al., 2009) and empirical results from zircon-like, zero-damage orthophosphates (Farley, 2007) that attempt to place an approximation on He diffusion in a pristine zircon. How do the author's new extrapolations match up to these results as a comparison? Finally, as Ginster et al. (2019) observed, it is very unlikely that a natural zircon would return to a pristine damage state following damage accumulation, so the likelihood of encountering a natural, zero-damage zircon is remote. I would like to see the authors acknowledge some of these points and address these questions. I think the authors have illuminated a valid and important point: that the ZRDAAM as defined by Guenthner et al. 2013 needs to be recalibrated at the low damage end and uncertainty needs to be accounted for. However, as I've hopefully conveyed, I would place the focus more upon using an endmember for which we can obtain kinetic information, rather than (as was done in Guenthner et al., 2013 and is done here) an idealized, extrapolated endmember that does not likely exist in nature.

*We have discussed this point above because we feel like this is the main point of the discussion. We have also addressed this in the text by reiterating that we are not aiming to produce a new model. We just want to propagate uncertainty.*

Second, the authors highlight a data set from Minnesota as an example with which to test their newly constrained ZRDAAM. My concern with this modeling setup, as articulated in a line specific comment below, is that an apples-to-apples comparison to the McDannell et al. 2022 study would include many more iterations. In that study, 500,000 burn-in, and 500,000 post were used, whereas here it appears that only 100,000 burn-in and 100,000 post are used. That is, the higher amount of uncertainty as to when Neoproterozoic cooling initiates could be that the model was not run for long enough.

*This is one reason why the models would be different. But there are many other reasons. We are not comparing the models to the McDannell model, but rather the results with different model parameters.*

Third, the authors call for additional diffusion experiments and new data to be added to the ZRDAAM, which I certainly agree with. I would point out though that some of these data have already been collected by Ginster and are published in her PhD dissertation, available through multiple repositories, including the University of Arizona library. This work is yet to be published in a peer-reviewed manuscript, but the data essentially agree with the Guenthner et al. (2013) diffusion data, which suggests that the variation (and therefore the uncertainty) in measurable diffusion kinetics is lower than for extrapolated endmembers. The point should be emphasized that, from an experimental perspective, the ZRDAAM remains on solid footing, although perhaps the endmember kinetics need to be recast, as my previous comment suggests.

*Totally agree with this point. Again, we just want to propagate uncertainty. However, we don't think it is appropriate to include the data in the inversion for the diffusion kinetics. These data are unpublished and we would rather wait until Ginster publishes her thesis. Of course, we can redo the calibration again.*

Line specific comments:

Line 73: Should acknowledge here that alternatives using an arguably more direct measure of alpha dose (Raman spectroscopy) have been proposed (Ginster et al. 2019) and incorporated into ZRDAAM (Guenthner, 2021). I would also elaborate the discussion here to comment on the different styles of damage that may influence He diffusion kinetics (i.e. alpha ejection, alpha recoil, fission track). The diffusion kinetics are calibrated to alpha dose (essentially alpha recoil), but the mode of annealing is debated and as yet not fully resolved. The Ginster et al. (2019) data and the model demonstration by Guenthner (2021) of these data are particularly salient given my comments above concerning an idealized zero-damage endmember.

*We have attempted to keep the paper simple and focus on a single message. Already, this paper is quite complicated.*

Line 85: An additional explanation here is that, as McDannell et al. 2022 showed, these models needed to be run for many more path iterations (at least 500,000 pre and post burn-in). In the Thurston et al. (2022) study, we ran models for only 100,000 pre and post burn-in, which was admittedly likely not enough. To this point, we have re-run some of these models with the greater number of paths (currently unpublished) and indeed the earlier portions of the time-temperature history remain under-constrained by zircon He (as discussed in Thurston et al., 2022). However, the late Miocene cooling remains robust.

*We have added that parameter space might not have been sampled sufficiently.*

Line 99: The phrase "varies by hundreds of millions of years" should be qualified here. As you suggest later in the manuscript, the variation is dependent on the number of grain inputs you have and the spread in date-eU space of those inputs. Moreover, my understanding (this could be clearer, see next comment below at line 238) is that this is for scenarios that incorporate the 2 sigma from the full kinetic distribution.

*This has been removed.*

Line 238: The focus in this paragraph on fixed endmembers seems out of place with one theme of this manuscript: kinetic uncertainties should be sampled in the rjMCMC approach. Some of this could be my confusion: am I correct that the models were run with Ea and D0 values that represent the 2 sigma of the kinetic distribution? If I'm not correct, then please more thoroughly explain how or why these specific kinetics were selected. If I am correct, then I understand that the authors are perhaps trying to show the worst-case, 2 sigma extremes from their distribution, but why not incorporate the full probability distribution as shown in figure 2 and sample that? The authors mention further below that computation limitations prohibit this exercise, but much of the discussion and implications seem to be

cast in light of the highest possible amount of variation. If the modeling incorporates the full distribution (and samples it) is the situation as dire? I am really curious to see the outcome of a modified MCMC proposal algorithm with a selection statement that samples the kinetic distribution.

*It is not clear to us how to do this at present. We have a produced a new version of QTQt that explicitly samples from our inferred model parameters. In this way the model does not attempt to find the best combination of diffusion parameters – the diffusion experiments are the best way to do this. Instead it incorporates this uncertainty into the analysis.*

Line 249: For a better apples-to-apples comparison here with the McDannell et al. (2022) study, 500k pre and 500k post burn-in is needed. As we have seen (and learned, see the comment about Thurston up above) these deep-time problems require at a minimum 1,000,000 path iterations.

*We tested this point and ran models with more and fewer iterations in the burn in phase. This doesn't seem to be a big problem if the chain is properly sampling parameter space.*

Line 294: Are the authors suggesting here that the binning and averaging approach has limitations? The point I'm most struck by, as the authors allude to, is that binning and averaging can eliminate the sensitivity of the whole data set by removing portions of date-eU space.

*We agree that this is problematic and would suggest not binning and averaging data. Instead, the distribution of age as a function of eU could be mapped out more completely.*

Willy Guenthner

UIUC

Citation: https://doi.org/10.5194/gchron-2022-23-RC5